# Inversion-based Latent Bayesian Optimization

**Jaewon Chu**,[*] **Jinyoung Park**,[*] **Seunghun Lee,** **Hyunwoo J. Kim**[†]
Computer Science & Engineering
Korea University
{allonsy07, lpmn678, llsshh319, hyunwoojkim}@korea.ac.kr

## Abstract

Latent Bayesian optimization (LBO) approaches have successfully adopted Bayesian optimization over a continuous latent space by employing an encoder-decoder architecture to address the challenge of optimization in a high dimensional or discrete input space. LBO learns a surrogate model to approximate the black-box objective function in the latent space. However, we observed that most LBO methods suffer from the 'misalignment problem', which is induced by the reconstruction error of the encoder-decoder architecture. It hinders learning an accurate surrogate model and generating high-quality solutions. In addition, several trust region-based LBO methods select the anchor, the center of the trust region, based solely on the objective function value without considering the trust region's potential to enhance the optimization process. To address these issues, we propose **Inv**ersion-based Latent **B**ayesian **O**ptimization (InvBO), a plug-and-play module for LBO. InvBO consists of two components: an inversion method and a potential-aware trust region anchor selection. The inversion method searches the latent code that completely reconstructs the given target data. The potential-aware trust region anchor selection considers the potential capability of the trust region for better local optimization. Experimental results demonstrate the effectiveness of InvBO on nine real-world benchmarks, such as molecule design and arithmetic expression fitting tasks. Code is available at https://github.com/mlvlab/InvBO.

## 1 Introduction

Bayesian optimization (BO) has been used in a wide range of applications such as material science [1], chemical design [2, 3], ,and hyperparameter optimization [4, 5]. The main idea of BO is probabilistically estimating the expensive black-box objective function using a surrogate model to find the optimal solution with minimum objective function evaluation. While BO has shown its success on continuous domains, applying BO over discrete input space is challenging [6, 7]. To address it, Latent Bayesian Optimization (LBO) has been proposed [8–14]. LBO performs BO over a latent space by mapping the discrete input space into the continual latent space with generative models such as Variational Auto Encoders (VAE) [15], consisting of an encoder $q_\phi$ and a decoder $p_\theta$. Unlike the standard BO, the surrogate model in LBO associates a latent vector $\mathbf{z}$ with an objective function value by emulating the composition of the objective function and the decoder of VAE.

In LBO, however, the reconstruction error of the VAE often leads to one latent vector $\mathbf{z}$ being associated with two different objective function values as explained in Figure 1. We observe that the discrepancy between $y$ and $y'$ (or $\mathbf{x}$ and $\mathbf{x}'$) hinders learning an accurate surrogate model $g$ and generating high-quality solutions. We name this the 'misalignment problem'. Most prior works [9, 11, 12] use the surrogate model $g^{\text{enc}}$, which is trained with the encoder triplet $(\mathbf{x}, \mathbf{z}, y)$, and

---

[*]equal contributions
[†]Corresponding author

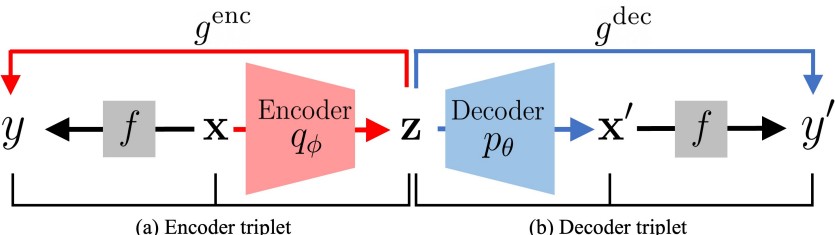

(a) Encoder triplet         (b) Decoder triplet

Figure 1: **Misalignment problem.** In LBO, a latent vector $\mathbf{z}$ can be associated with two function values $y$ and $y'$ due to the reconstruction error of the VAE, *i.e.*, $\mathbf{x} \neq \mathbf{x}'$. (a) In Encoder triplet $(\mathbf{x}, \mathbf{z}, y)$, latent vector $\mathbf{z}$ is associated with $f(\mathbf{x})$, where $\mathbf{x}$ is the original input to the encoder, *i.e.*, $\mathbf{z} = q_\phi(\mathbf{x})$. (b) In Decoder triplet $(\mathbf{x}', \mathbf{z}, y')$, $\mathbf{z}$ is associated with $y' = f(\mathbf{x}')$, which is the objective function value of reconstructed input value $\mathbf{x}'$ using the decoder, *i.e.*, $\mathbf{x}' = p_\theta(\mathbf{z})$. The discrepancy between $y$ and $y'$ hinders learning the accurate surrogate model $g$. We name this the 'misalignment problem'.

generate solutions $\mathbf{x}'$ via decoder $p_\theta$. Since $g^{\text{enc}}$ fails to estimate the composite function of $p_\theta$ and $f$, this approach often results in suboptimal outcomes. Some works [13, 14] handle the misalignment problem by employing the surrogate model $g^{\text{dec}}$ trained with the decoder triplet $(\mathbf{x}', \mathbf{z}, y')$. However, they request a huge amount of additional oracle calls to obtain the decoder triplet, which leads to inefficient optimization. In addition, several existing LBO methods [13, 14, 16] adopt the trust region method to restrict the search space and have shown performance gain. Most prior works select the anchor, the center of the trust region, as the current optimal point. This objective function value-based anchor selection overlooks the potential to benefit the optimization performance of the latent vectors within the trust region.

In this work, we propose an **Inv**ersion-based Latent **B**ayesian **O**ptimization (**InvBO**), a plug-and-play module for VAE-based LBO methods. InvBO consists of two components: the inversion method and a potential-aware trust region anchor selection. The inversion method addresses the misalignment problem by inverting decoder $p_\theta$ to find the latent code that yields $\mathbf{x}$ without any additional oracle call. We theoretically analyze that our inversion method decreases the upper bound of the error between the surrogate model and the objective function within the trust region. The potential-aware trust region anchor selection method selects the anchor considering not only the observed objective function value but also the potential to enhance the optimization process of the latent vectors that the trust region contains. We provide the experimental evaluation on nine different tasks, Guacamol, DRD3, and arithmetic expression fitting task to show the general effectiveness of InvBO. Specifically, plug-and-play results of InvBO over diverse prior LBO works show a large performance gain and achieved state-of-the-art performance.

The contributions of our paper are as follows:

- We propose the inversion method to address the misalignment problem in LBO by generating the decoder triplet without using any additional oracle calls.

- We propose the potential-aware trust region anchor selection, aiming to select the centers of trust regions considering the latent vectors expected to benefit the optimization process within the trust regions.

- By combining the inversion method and potential-aware trust region anchor selection, we propose Inversion-based Latent Bayesian Optimization (InvBO), a novel plug-and-play module for LBO, and achieve state-of-the-art performance on the nine different tasks.

## 2   Related Works

### 2.1   Latent Bayesian Optimization

The goal of Latent Bayesian Optimization (LBO) [8, 9, 11–14, 16–19] is to learn a latent space to enable optimization over a continuous space from discrete or structured input (*e.g.,* graph or image). LBO consists of a Variational AutoEncoder (VAE) to generate data from the latent representation and a surrogate model (*e.g.*, Gaussian process) to map the latent representation into the objective score. Some works on the LBO have designed new decoder architectures [8, 20–23] to perform the

reconstruction better, while other works have proposed learning mechanisms [9–14, 17] to alleviate the discrepancy between the latent space and input space. LOL-BO [13] adapts the concept of trust-region to the latent space and jointly learns a VAE and a surrogate model to search the data point in the local region. CoBO [14] designs new loss to encourage the correlation between the distances in the latent space and objective function.

## 2.2 Inversion in Generative Models

Inversion has widely been applied to a variety of generative models such as Generative Adversarial Networks (GANs) [24, 25] and Diffusion models [26–29]. Inversion is the process of finding the latent code $\mathbf{z}_{\text{inv}}$ of a given image to manipulate images with generative models. Formally, given an image $\mathbf{x}$ and the well-trained generator $G$, the inversion can be written as:

$$\mathbf{z}_{\text{inv}} = \arg\min_{\mathbf{z} \in \mathcal{Z}} d_{\mathcal{X}}(G(\mathbf{z}), \mathbf{x}), \tag{1}$$

where $d_{\mathcal{X}}(\cdot, \cdot)$ denotes the distance metric in the image space $\mathcal{X}$, and $\mathcal{Z}$ is the latent space. To solve Eq. (1), most inversion-based works can be generally classified as two approaches: optimization-based and learning-based methods. The optimization-based inversion [30–33] iteratively finds a latent vector to reconstruct the target image $\mathbf{x}$ through the fixed generator. The learning-based inversion [25, 34, 35] trains the encoder for mapping the image $\mathbf{x}$ to the latent code $\mathbf{z}$ while fixing the decoder. In this work, we introduce the concept of inversion to find the latent vector that can generate a desired sample for constructing an aligned triplet and we use the optimization-based inversion.

## 3 Preliminaries

**Bayesian optimization.** Bayesian optimization (BO) is a powerful and sample-efficient optimization algorithm that aims at searching the input $\mathbf{x}$ with a maximum objective value $f(\mathbf{x})$, which is formulated as:

$$\mathbf{x}^* = \arg\max_{\mathbf{x} \in \mathcal{X}} f(\mathbf{x}), \tag{2}$$

where the black-box objective function $f : \mathcal{X} \mapsto \mathcal{Y}$ is assumed expensive to evaluate, and $\mathcal{X}$ is a feasible set. Since the objective function $f$ is unknown or cost-expensive, BO methods probabilistically emulate the objective function by a surrogate model $g$ with observed dataset $\mathcal{D} = \{(\mathbf{x}^i, y^i) | y^i = f(\mathbf{x}^i)\}_{i=1}^n$. With the surrogate model $g$, the acquisition function $\alpha$ selects the most promising point $\mathbf{x}^{n+1}$ as the next evaluation point while balancing exploration and exploitation. BO repeats this process until the oracle budget is exhausted.

**Trust region-based local Bayesian optimization.** Classical Bayesian optimization methods often suffer from the difficulty of the optimization in a high dimensional space [36]. To address this problem, TuRBO [36] adopts trust regions to limit the search space to small regions. The anchor (center) of trust region $\mathcal{T}$ is selected as a current optimal point, and the size of the trust region is scheduled during the optimization process. At the beginning of the optimization, the side length of all trust regions is set to $L_{\text{init}}$. When the trust region $\mathcal{T}$ updates the best score $\tau_{\text{succ}}$ times in a row, the side length becomes twice until it reaches $L_{\text{max}}$. Similarly, when it fails to update the best score $\tau_{\text{fail}}$ times in a row, the side length becomes half. When $L$ falls below a $L_{\text{min}}$, the side length of the trust region is set to $L_{\text{init}}$ and restart the scheduling. Recently, LOL-BO [13] adapted trust region-based local optimization to LBO, and has shown performance gain.

## 4 Method

In this section, we present an Inversion-based Latent Bayesian Optimization (InvBO) consisting of an inversion and a novel trust region anchor selection method for effective and efficient optimization. We first describe latent Bayesian optimization and the misalignment problem of it (Section 4.1). Then, we introduce the inversion method to address the misalignment problem without using any additional oracle budgets (Section 4.2). Lastly, we present a potential-aware trust region anchor selection for better local search space (Section 4.3).

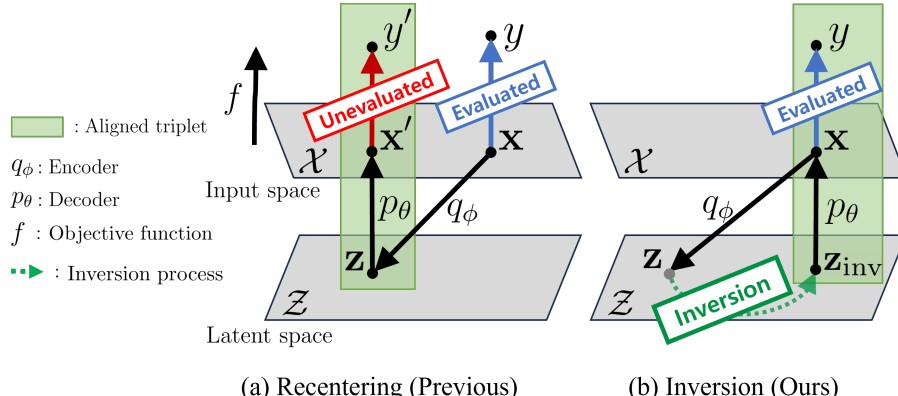

(a) Recentering (Previous)    (b) Inversion (Ours)

Figure 2: **Comparison of solutions to the misalignment problem.** (a) Some works [13, 14] solve the misalignment problem by the recentering technique that generates the aligned triplet $(\mathbf{x}', \mathbf{z}, y')$. However, it requests additional oracle calls as $y' = f(\mathbf{x}')$ is unevaluated, and does not fully use the evaluated function value $y = f(\mathbf{x})$. (b) The inversion method (ours) aims to find $\mathbf{z}_{\mathrm{inv}}$ that generates the evaluated data $\mathbf{x}$ to get the aligned triplet $(\mathbf{x}, \mathbf{z}_{\mathrm{inv}}, y)$ without any additional oracle calls.

## 4.1 Misalignment in Latent Bayesian Optimization

BO has proven its effectiveness in various areas where input space $\mathcal{X}$ is continuous, however, BO over the discrete domain, such as chemical design, is a challenging problem. To handle this problem, VAE-based latent Bayesian optimization (LBO) has been proposed [8, 11, 13, 14] that leverages BO over a continuous space by mapping the discrete input space $\mathcal{X}$ to a continuous latent space $\mathcal{Z}$. Variational autoencoder (VAE) is composed of encoder $q_\phi : \mathcal{X} \mapsto \mathcal{Z}$ to compute the latent representation $\mathbf{z}$ of the input data $\mathbf{x}$ and decoder $p_\theta : \mathcal{Z} \mapsto \mathcal{X}$ to generate the data $\mathbf{x}$ from the latent $\mathbf{z}$.

Given the objective function $f$, latent Bayesian optimization can be formulated as:

$$\mathbf{z}^* = \arg\max_{\mathbf{z} \in \mathcal{Z}} f(p_\theta(\mathbf{z})), \tag{3}$$

where $p_\theta(\mathbf{z})$ is a generated data with the decoder $p_\theta$ and $\mathcal{Z}$ is a latent space. Unlike the standard BO, the surrogate model $g$ aims to emulate the function $f \circ p_\theta : \mathcal{Z} \mapsto \mathcal{Y}$. To the end, the surrogate model is trained with aligned dataset $\mathcal{D} = \{(\mathbf{x}^i, \mathbf{z}^i, y^i)\}_{i=1}^n$, where $\mathbf{x}^i = p_\theta(\mathbf{z}^i)$ is generated by the decoder $p_\theta : \mathcal{Z} \mapsto \mathcal{X}$ and $y^i = f(\mathbf{x}^i)$ is the objective value of $\mathbf{x}^i$ evaluated via the black box objective function $f : \mathcal{X} \mapsto \mathcal{Y}$. In the rest of our paper, we define that the dataset is aligned when all triplets satisfy the above conditions (*i.e.,* all triplets are the decoder triplets explained in Figure 1), and the dataset is misaligned otherwise. We define the 'misalignment problem' as the misaligned dataset hinders the accurate learning of the surrogate model $g$.

Most existing LBO works [11–14] overlook the misalignment problem 1, which originates from two processes: (i) construction of initial dataset $\mathcal{D}^0$ and (ii) update of VAE.

**Construction of initial dataset $\mathcal{D}^0$.** Since initial dataset $\mathcal{D}^0$ is composed of pairs of input data and its corresponding objective value $\{(\mathbf{x}^i, y^i) | y^i = f(\mathbf{x}^i)\}_{i=1}^n$, LBO requires latent vectors $\{\mathbf{z}^i\}_{i=1}^n$ to train the surrogate model. Most works compute a latent vector $\mathbf{z}^i$ as $\mathbf{z}^i = q_\phi(\mathbf{x}^i)$ under the assumption that VAE completely reconstructs every data points (*i.e.,* $p_\theta(q_\phi(\mathbf{x}^i)) = \mathbf{x}^i$), which is difficult to be satisfied in every case. This results in the data misalignment ($\mathbf{x}^i \neq p_\theta(\mathbf{z}^i)$) during the construction of initial dataset $\mathcal{D}^0$.

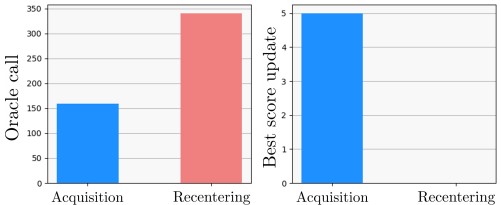

Figure 3: (Left) The number of oracle calls to evaluate the queries selected by the acquisition function (blue) and during the recentering (Red). (Right) The number of objective function evaluation that updates the best score.

**Update of VAE.** In LBO works, updating VAE during the optimization plays a crucial role in adapting well to newly generated samples. However, due to the update of VAE, the previously generated triplet $(\mathbf{x}, \mathbf{z}, y)$ cannot ensure the alignment since $\mathbf{z}$ was computed by the VAE before the update. Previous LBO works [13, 14] solve the misalignment problem originating from the VAE update with a recentering technique that requests additional oracle calls to generate the aligned dataset

$\mathcal{D} = \{(\mathbf{x}'^i, q_\phi(\mathbf{x}^i), f(\mathbf{x}'^i))\}_{i=1}^n$ where $\mathbf{x}'^i = p_\theta(q_\phi(\mathbf{x}^i))$ as shown in Figure 2 (Left). But, it has a limitation to consuming a huge amount of additional oracle calls while they do not update the best score (Figure 3). Note that prior works do not explicitly mention the additional oracle calls during the recentering technique, but it can be verified by the official GitHub code. Further details about oracle consumption of the recentering are provided in the supplement Section H.

## 4.2 Inversion-based Latent Bayesian Optimization

Our primary goal is training the surrogate model $g$ to correctly emulate the composite function $f \circ p_\theta : \mathcal{Z} \mapsto \mathcal{Y}$ via constructing an aligned dataset without consuming additional oracle calls. To the end, we propose an inversion method that inverts the target discrete data $\mathbf{x}$ into the latent vector $\mathbf{z}$ that satisfies $\mathbf{x} = p_\theta(\mathbf{z})$ for dataset alignment as shown in Figure 2 (Right). With a pre-trained frozen decoder $p_\theta$, the latent vector $\mathbf{z}$ can be optimized by:

$$\mathbf{z}_{\text{inv}} = \arg\min_{\mathbf{z} \in \mathcal{Z}} d_{\mathcal{X}}(\mathbf{x}, p_\theta(\mathbf{z})), \tag{4}$$

where $\mathbf{x}$ is a target data and $d_{\mathcal{X}}$ is a distance function in the input space $\mathcal{X}$. We use the normalized Levenshtein distance [37] as our distance function, $d_{\mathcal{X}}$, which can be applied to any string-form data. Our inversion method, however, is flexible and can utilize any task-specific distance functions, such as Tanimoto similarity [38] for molecule design tasks. To solve the Eq. (4), we iteratively update a latent vector $\mathbf{z}$ to find $\mathbf{z}_{\text{inv}}$ that reconstructs the target data $\mathbf{x}$. We provide the overall pseudocode of the inversion method in Algorithm 1.

The initialization strategy of latent vector $\mathbf{z}$ plays a key role in the optimization-based inversion process. We set the initialization point of latent vector $\mathbf{z}$ as an output of a pre-trained encoder $q_\phi(\mathbf{x})$ given target discrete data $\mathbf{x}$ as in line 1. We iteratively update the latent vector $\mathbf{z}$ with the cross-entropy loss used in VAE training in line 3 until it reaches the maximum number of iterations $T$. Before the iteration budget is exhausted, we finish the inversion process when the distance between the generated data $p_\theta(\mathbf{z})$ and target data $\mathbf{x}$ is less than $\epsilon$ as our goal is finding the latent vector $\mathbf{z}_{\text{inv}}$ that satisfies Eq. (4), which is denoted in line 4. The inversion method

---

**Algorithm 1** Inversion

**Input:** Encoder $q_\phi$, decoder $p_\theta$, target data $\mathbf{x}$, max iteration $T$, distance function $d_{\mathcal{X}}$, learning rate $\eta$, reconstruction loss $\mathcal{L}$

1: Initialize $\mathbf{z}^{(0)} \leftarrow q_\phi(\mathbf{x})$
2: **for** $t = 0, 1, ..., T - 1$ **do**
3:      $\mathbf{z}^{(t+1)} \leftarrow \mathbf{z}^{(t)} - \eta \nabla_{\mathbf{z}^{(t)}} \mathcal{L}\left(p_\theta(\mathbf{z}^{(t)}), \mathbf{x}\right)$
4:      **if** $d_{\mathcal{X}}(\mathbf{x}, p_\theta(\mathbf{z}^{(t+1)})) < \epsilon$ **then**    $\triangleright$ *Eq.* (4)
5:          **return** $\mathbf{z}^{(t+1)}$
6:      **end if**
7: **end for**
8: **return** $\mathbf{z}^{(T)}$

---

generates the aligned dataset during the construction of the initial dataset $\mathcal{D}^0$ and the update of VAE to handle the misalignment problem.

We theoretically show that optimizing the latent vector $\mathbf{z}$ to satisfy $d_{\mathcal{X}}(\mathbf{x}, p_\theta(\mathbf{z})) \approx 0$ with inversion plays a crucial role in minimizing the upper bound of the error between the posterior mean of the surrogate model and the objective function value within the trust region centered at $\mathbf{z}$.

**Proposition 1.** *Let $f$ be a black-box objective function and $m$ be a posterior mean of Gaussian process, $p_\theta$ be a decoder of the variational autoencoder, $c$ be an arbitrarily small constant, $d_{\mathcal{X}}$ and $d_{\mathcal{Z}}$ be the distance function on input $\mathcal{X}$ and latent $\mathcal{Z}$ spaces, respectively. The distance function $d_{\mathcal{X}}$ is bounded between 0 and 1, inclusive. We assume that $f$, $m$ and the composite function of $f$ and $p_\theta$ are $L_1$, $L_2$, and $L_3$-Lipschitz continuous functions, respectively. Suppose the following assumptions are satisfied:*

$$\begin{aligned} |f(\mathbf{x}) - m(\mathbf{z})| &\leq c, \\ d_{\mathcal{X}}(\mathbf{x}, p_\theta(\mathbf{z})) &\leq \gamma. \end{aligned} \tag{5}$$

*Then the difference between the posterior mean of the arbitrary point $\mathbf{z}'$ in the trust region centered at $\mathbf{z}$ with trust radius $\delta$ and the black box objective value is upper bounded as:*

$$|f(p_\theta(\mathbf{z}')) - m(\mathbf{z}')| \leq c + \gamma \cdot L_1 + \delta \cdot (L_2 + L_3), \tag{6}$$

*where $d_{\mathcal{Z}}(\mathbf{z}, \mathbf{z}') \leq \delta$.*

The proof is available in Section A. We assume that the black box function $f$, the posterior mean of Gaussian process $m$, and the objective function $f \circ p_\theta$ are Lipschitz continuous functions, which is a

common assumption in Bayesian optimization [39, 40] or global optimization [41]. Proposition 1 implies that the upper bound of the Gaussian process prediction error within the trust region can be minimized by reducing the distance between $\mathbf{x}$ and $p_\theta(\mathbf{z})$, denoted as $\gamma$. Our inversion method reduces $\gamma$ by generating an aligned dataset without additional oracle calls. In Eq. (6), the constant $c$ represents the accuracy of the surrogate model, which can be improved during training. Some LBO works [12, 13] introduce regularization terms to learn a smooth latent space, implicitly reducing $L_3$, the Lipschitz constant of the composite function $f \circ p_\theta$. CoBO [14] further proposes regularization losses that explicitly reduce $L_3$ with theoretical grounding. Since the surrogate model $m$ emulates the composite function $f \circ p_\theta$, its Lipschitz constant $L_2$ can also be reduced along with $L_3$.

### 4.3 Potential-aware Trust Region Anchor Selection

Here, we propose a potential-aware trust region anchor selection to consider both the objective function value $y$ and the potential ability of the trust region to benefit the optimization. Previous trust region-based BO works [13, 14, 36, 42] select the anchor based on the corresponding objective function value only. However, this objective score-based anchor selection does not consider that the trust region contains latent points expected to enhance the optimization performance.

We design a potential score to measure the potential ability to enhance the optimization process of the trust region. To compute it, we use the acquisition function value, which is generally employed to measure the potential ability to improve the optimization process of a given data point. Specifically, we employ Thompson Sampling, a well-established acquisition function used in previous trust region-based methods [13, 14, 36]. Formally, the potential score of each trust region $\mathcal{T}_i$ is computed as follows:

$$\alpha_{\text{pot}}^i = \max_{\mathbf{z} \in Z_{\text{cand}}^i} \hat{f}(\mathbf{z}) \text{ where } \hat{f} \sim \mathcal{GP}\left(\mu(\mathbf{z}), k(\mathbf{z}, \mathbf{z}')\right), \tag{7}$$

where $i$ is an index of the candidate anchor point, $Z_{\text{cand}}^i$ is the candidate set sampled from the trust region $\mathcal{T}_i$ and $\hat{f}$ is a sampled function from the surrogate model (*e.g.*, GP) posterior.

As the scale of objective function values $Y = \{y^1, y^2, ..., y^n\}$ and that of the potential ability of each trust region $A = \{\alpha_{\text{pot}}^1, \alpha_{\text{pot}}^2, ..., \alpha_{\text{pot}}^n\}$ is changed dynamically during the optimization process, we calculate a scaled potential score $\alpha_{\text{pot}}^i$ by adjusting the scale of $\alpha_{\text{max}}^i$ according to the $Y$:

$$\alpha_{\text{scaled}}^i = \frac{\alpha_{\text{pot}}^i - A_{\text{min}}}{A_{\text{max}} - A_{\text{min}}} \times (Y_{\text{max}} - Y_{\text{min}}), \tag{8}$$

where $A_{\text{max}} = \max_i \left[\alpha_{\text{pot}}^i\right]$ and $A_{\text{min}} = \min_i \left[\alpha_{\text{pot}}^i\right]$ denote the maximum and minimum value of $A$, respectively. $Y_{\text{max}} = \max_i \left[y^i\right]$ and $Y_{\text{min}} = \min_i \left[y^i\right]$ indicate the maximum and minimum value of $Y$, respectively. Based on the scaled potential score $\alpha_{\text{scaled}}^i$, the final score $s^i$ of each anchor is calculated as:

$$s^i = y^i + \alpha_{\text{scaled}}^i. \tag{9}$$

Our final score takes into account the objective function value of the anchor (observed value) and the potential score (model's prediction) for the better local search space. Finally, we select the anchors with the highest final score $s^i$. We summarize our potential-aware trust region anchor selection schema in Algorithm 2 of Section N.

## 5 Experiments

### 5.1 Tasks

We measure the performance of the proposed method named InvBO on nine different tasks with three Bayesian optimization benchmarks: Guacamol [43], DRD3, and arithmetic expression fitting tasks [6, 8, 11–13, 44]. Guacamol and the DRD3 benchmark tasks aim to find molecules with the most necessary properties. For Guacamol benchmarks, we use seven challenging tasks, Median molecules 2 (med2), Zaleplon MPO (zale), Perindopril MPO (pdop), Amlodipine MPO (adip), Osimertinib MPO (osmb), Ranolazine MPO (rano), and Valsartan SMARTS (valt). To show the effectiveness of our InvBO in various settings, we also conducted experiments in a large budget setting, which is used in previous works [13, 14]. The goal of the arithmetic expression fitting task is to generate single-variable expressions that minimize the distance from a target expression (*e.g.*, $1/3 + x + \sin(x \times x)$). More details of each benchmark are provided in Section K.

Table 1: Optimization results of applying InvBO to several trust region-based LBOs on Guacamol benchmark tasks. A higher score is a better one.

| Task | Median molecules 2 (med2) | | | Valsartan SMARTS (valt) | | |
|---|---|---|---|---|---|---|
| Num Oracle | 100 | 300 | 500 | 100 | 300 | 500 |
| TuRBO-$L$ | 0.186±0.000 | 0.186±0.000 | 0.186±0.000 | 0.000±0.000 | 0.000±0.000 | 0.000±0.000 |
| TuRBO-$L$ + InvBO | 0.186±0.000 | **0.194±0.002** | **0.202±0.001** | 0.000±0.000 | **0.024±0.017** | **0.212±0.092** |
| LOL-BO | 0.186±0.000 | 0.186±0.000 | 0.190±0.001 | 0.000±0.000 | 0.000±0.000 | 0.000±0.000 |
| LOL-BO + InvBO | **0.189±0.002** | **0.204±0.005** | **0.227±0.010** | 0.000±0.000 | **0.007±0.005** | **0.171±0.039** |
| CoBO | 0.186±0.000 | 0.188±0.002 | 0.191±0.003 | 0.000±0.000 | 0.000±0.000 | 0.000±0.000 |
| CoBO + InvBO | **0.187±0.001** | **0.203±0.004** | **0.214±0.006** | 0.000±0.000 | **0.042±0.013** | **0.348±0.107** |

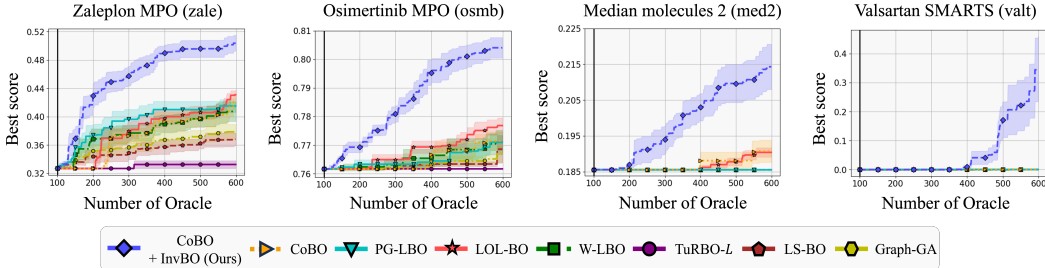

Figure 4: Optimization results on Guacamol benchmark tasks. The lines and ranges indicate the average and standard error of ten runs under the same settings. A higher score is a better score.

## 5.2 Baselines

We compare to six latent Bayesian Optimization methods: LS-BO, TuRBO [36], W-LBO [11], LOL-BO [13], CoBO [14], and PG-LBO [17]. We also compare with GB-GA [45], a widely used genetic algorithm for graph structure data. LOL-BO and CoBO employ the decoder triplet generated by the recentering technique, and the other baselines employ the encoder triplet during the optimization process. In the case of LOL-BO and CoBO, we substitute the recentering technique with the inversion method. We provide the details of each baseline in Section L.

## 5.3 Implementation Details

For the arithmetic expression fitting task, we follow other works [8, 11, 13, 14] to employ Grammar-VAE model [8]. For the *de novo* molecule design tasks such as Guacamol benchmark and DRD3 tasks, we use SELFIES VAE [13] following recent works [13, 14]. In all tasks, we adopt sparse variational GP [46] with deep kernel [47] as our surrogate model. In the inversion method, the learning rate is set to 0.1 in all experiments. The further implementation details are provided in Section M.

## 5.4 Experimental Results

We apply our InvBO to several trust region-based LBOs, TuRBO-$L$, LOL-BO [13] and CoBO [14], and provide optimization results on two Guacamol benchmark tasks, med2 and valt. All results are average scores of ten runs under the identical settings. Table 1 demonstrates that our InvBO consistently improves all LBO models in two tasks by a large margin. In particular, in the valt task, all baseline models with InvBO demonstrate significant performance improvements and CoBO with InvBO achieves a 0.348 score gain while the baseline models without InvBO fail in optimization. More results of other baselines with InvBO on other tasks are provided in Section E.

Figure 4 provides the optimization results on four Guacamol benchmark tasks including med2, zale, osmb, and valt. Each subfigure shows the number of evaluations of the black box objective function (Number of Oracle) and the corresponding average and standard error of the objective score (Best Score). Our InvBO built on CoBO achieves the best performance in all four tasks. Further results of other tasks are provided in Section F.

We also conduct experiments to demonstrate the effectiveness of our InvBO on DRD3 and arithmetic expression fitting tasks and large-budget settings. The experimental results are illustrated in Fig-

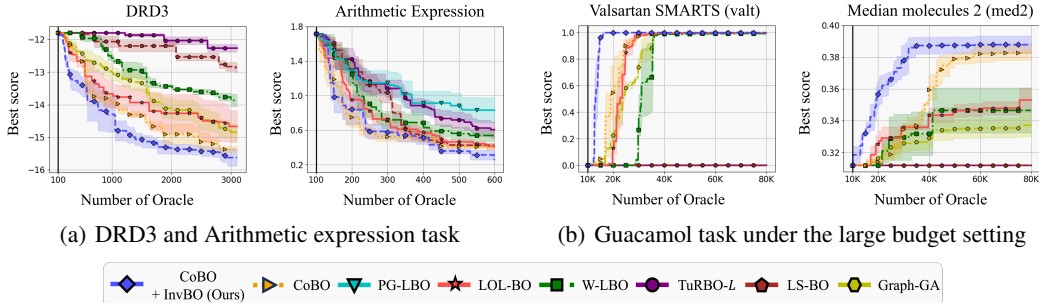

(a) DRD3 and Arithmetic expression task    (b) Guacamol task under the large budget setting

Figure 5: Optimization results on various tasks and settings. Note that: (a) A lower score is a better score. (b) A higher score is a better score.

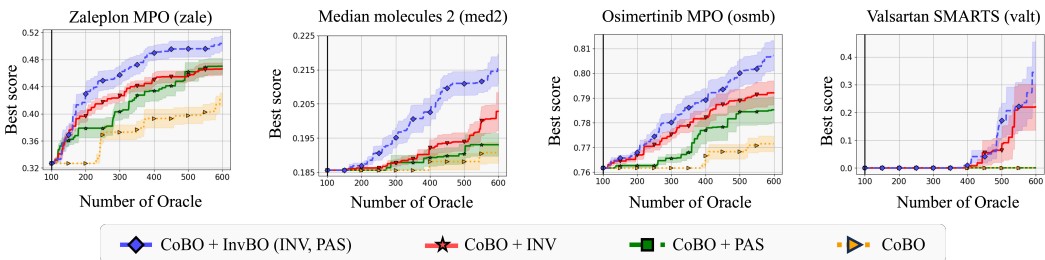

Figure 6: Optimization results of component ablation on zale, med2, osmb, and valt.

ure 5(a) and Figure 5(b), respectively. Please note that both DRD3 and arithmetic expression tasks aim to minimize the score while the goal of Guacamol tasks is increasing the score. Figure 5 shows that our InvBO applied to CoBO achieves the best performance. We provide further optimization results on other tasks under the large budget settings in Section G. These results demonstrate that InvBO is effective in diverse tasks and settings.

# 6 Analysis

## 6.1 Ablation Study

We conduct additional experiments to verify the contribution of each component in our InvBO: the inversion method (INV), and the potential-aware trust region anchor selection method (PAS). Figure 6 shows the optimization results of the ablation study on med2, zale, osmb, and valt tasks. From the figure, models with the inversion method (*i.e.,* CoBO with INV and InvBO) replace the recentering technique as the inversion method, while models without the inversion method (*i.e.,* vanilla CoBO and CoBO with PAS) employ the recentering technique. Notably, both components of our method contribute to the optimization process, and the combination work, InvBO, consistently obtains better objective scores compared to other models. Specifically, in osmb task, the average best score achieved by the methods with the PAS, INV and both (*i.e.,* InvBO) shows 0.784, 0.792, and 0.804 score gains compared to vanilla CoBO, respectively.

## 6.2 Analysis on Misalignment Problem and Inversion

To further prove that the misaligned dataset hinders the accurate learning of the surrogate model (*i.e.,* misalignment problem), we compare the performance of the surrogate model trained with aligned and misaligned datasets on the med2 task. Figure 7 shows the fitting results of the surrogate model trained with encoder triplets ($g^{\text{enc}}$, left) and decoder triplets ($g^{\text{dec}}$, right), respectively. Figure 7 demonstrates that the surrogate model $g^{\text{dec}}$ approximates the objective function accurately, while $g^{\text{enc}}$ fails to emulate the objective function. Further details of an experiment are provided in Section I.

In Figure 8, we compare the optimization results of CoBO using decoder triplets and encoder triplets on the valt and med2 tasks. Both models use the potential-aware anchor selection, and the decoder triplets are made by our inversion method. CoBO using the decoder triplets shows superior optimization performance over the CoBO using the encoder triplets on both tasks. This implies that

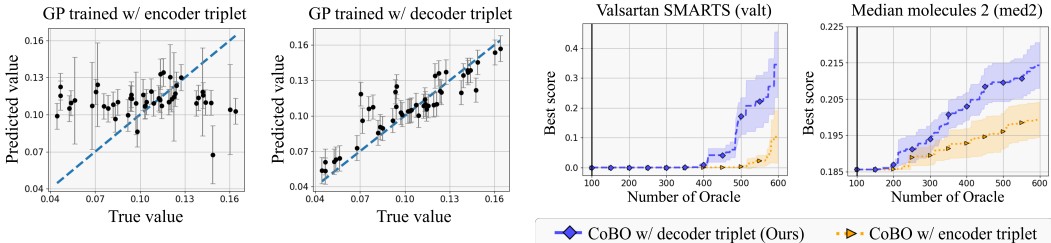

Figure 7: Gaussian process fitting results trained with encoder triplets and decoder triplets.

Figure 8: Optimization results using encoder triplet and decoder triplet on valt and med2 tasks.

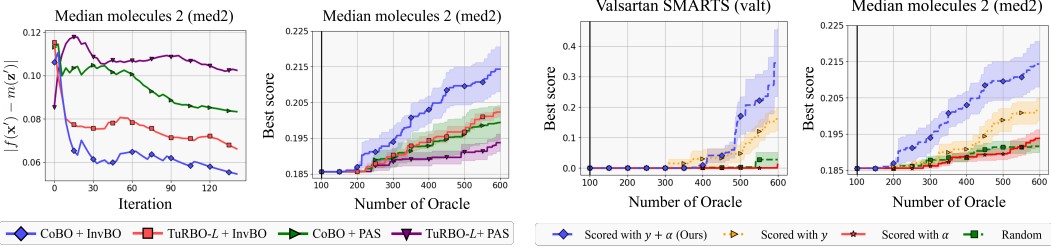

Figure 9: (Left) Gaussian process prediction error within the trust region. (Right) Optimization results on med2 tasks.

Figure 10: Optimization results with diverse trust region anchor selections on valt and med2 tasks.

the misaligned dataset hinders the accurate learning of the surrogate model, which leads to suboptimal optimization results, and the inversion method handles the problem by generating the decoder triplet.

### 6.3 Effects of Inversion on Proposition 1

In Section 4.2, we provide the upper bound of the error between the predicted and ground-truth objective value. In Eq. (6), the Lipschitz constant of the objective function $L_1$ and the trust region radius $\delta$ is fixed or the hyper-parameter and the constant $c$ can be improved by learning the surrogate model. In the end, we can reduce the upper bound of the objective value prediction error by reducing the three components: $\gamma$, $L_2$, and $L_3$, which implies the distance between $\mathbf{x}$ and $p_\theta(\mathbf{z})$, the Lipschitz constant of function $m$ and $f \circ p_\theta$, respectively. Our inversion method reduce $\gamma$ by searching the latent vector $\mathbf{z}_{\text{inv}}$ that satisfies $\mathbf{x} = p_\theta(\mathbf{z}_{\text{inv}})$. Previously, CoBO [14] proposed regularization losses that reduce $L_3$. Since the surrogate model emulates the composite function $f \circ p_\theta$, these regularization losses can reduce $L_2$ along with $L_3$.

Figure 9 (left) shows the regularization losses of CoBO and our inversion method reduces the objective value prediction error. CoBO-based models (*i.e.,* CoBO+PAS and CoBO+InvBO) employ the CoBO regularization losses, and models with InvBO (*i.e.,* TuRBO-$L$+InvBO and CoBO+InvBO) employ our inversion method. TuRBO-$L$ does not use the CoBO regularization losses nor our inversion method, and models with PAS (*i.e.,* TuRBO-$L$+PAS and CoBO+PAS) employ encoder triplets. Applying the regularization losses and our inversion method reduces the objective value prediction error, respectively, but our inversion method shows a larger error reduction. The combination of regularization losses and our inversion shows the smallest prediction error, which implies our inversion method robustly complements existing methods. We provide the optimization results of each model on the med2 task in Figure 9 (right). These results demonstrate that reducing the objective function prediction error plays a crucial role in optimization performance.

### 6.4 Comparing Diverse Anchor Selection Methods

To further prove the importance of our potential-aware anchor selection method, we perform BO with the diverse anchor selection methods: random, acquisition $\alpha$, and objective score $y$. Random indicates randomly selected anchors, and acquisition and objective indicate anchors are selected based on the max acquisition function value and objective score, respectively. All models use the inversion method, and the optimization results on valt and med2 tasks are in Figure 10. Ours and objective

score-based anchor selection rapidly find high-quality data compared to the random and acquisition-based selections. However, objective score-based anchor selection shows inferior performance than our potential-aware anchor selection. This indicates that both the uncertainty of the surrogate model and objective function value need to be considered for exploration and exploitation.

## 7   Conclusion

We propose Inversion-based Latent Bayesian Optimization (InvBO), a plug-and-play module for LBO. We introduce the inversion method that inverts the decoder to find the latent vector for generating the aligned dataset. Additionally, we present the potential-aware trust region anchor selection that considers not only the corresponding objective function value of the anchor but also the potential ability of the trust region. From our experimental results, InvBO achieves state-of-the-art performance on nine LBO benchmark tasks. We also theoretically demonstrate the effectiveness of our inversion method and provide a comprehensive analysis to show the effectiveness of our InvBO.

## Broader Impacts

One of the contributions of this paper is molecular design optimization, which requires careful consideration due to its unintentional applications such as the generation of toxic. We believe that our work has a lot of positive aspects to accelerate the development of chemical and drug discovery with an inversion-based latent Bayesian optimization method.

## Limitations

The performance of the methods proposed in the paper depends on the quality of the generative model. For example, if the objective function is related to the molecule property, the generated model such as the VAE should have the ability to generate the proper molecule to have good performance of optimization.

## Acknowledgement

This work was partly supported by ICT Creative Consilience Program through the Institute of Information & Communications Technology Planning & Evaluation (IITP) (IITP-2024-RS-2020-II201819, 10%) and the National Research Foundation of Korea (NRF) (NRF-2023R1A2C2005373, 45%) grant funded by the Korea government (MSIT), and the Virtual Engineering Platform Project (Grant No. P0022336, 45%), funded by the Ministry of Trade, Industry & Energy (MoTIE, South Korea).

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

## A A Proof of Proposition 1

In this section, we provide the proof of Proposition 1.

**Proposition 1.** *Let $f$ be a black-box objective function and $m$ be a posterior mean of Gaussian process, $p_\theta$ be a decoder of the variational autoencoder, $c$ be an arbitrarily small constant, $d_\mathcal{X}$ and $d_\mathcal{Z}$ be the distance function on input $\mathcal{X}$ and latent $\mathcal{Z}$ spaces, respectively. The distance function $d_\mathcal{X}$ is bounded between 0 and 1, inclusive. We assume that $f$, $m$ and the composite function of $f$ and $p_\theta$ are $L_1$, $L_2$, and $L_3$-Lipschitz continuous functions, respectively. Suppose the following assumptions are satisfied:*

$$|f(\mathbf{x}) - m(\mathbf{z})| \leq c,$$
$$d_\mathcal{X}(\mathbf{x}, p_\theta(\mathbf{z})) \leq \gamma. \tag{10}$$

*Then the difference between the posterior mean of the arbitrary point $\mathbf{z}'$ in the trust region centered at $\mathbf{z}$ with trust radius $\delta$ and the black box objective value is upper bounded as:*

$$|f(p_\theta(\mathbf{z}')) - m(\mathbf{z}')| \leq c + \gamma \cdot L_1 + \delta \cdot (L_2 + L_3), \tag{11}$$

*where $d_\mathcal{Z}(\mathbf{z}, \mathbf{z}') \leq \delta$.*

*Proof.* With $L$-Lipschitz continuity, we have:

$$|f(\mathbf{x}) - f(p_\theta(\mathbf{z}))| \leq L_1 \cdot d_\mathcal{X}(\mathbf{x}, p_\theta(\mathbf{z})) \leq \gamma \cdot L_1,$$
$$|m(\mathbf{z}) - m(\mathbf{z}')| \leq L_2 \cdot d_\mathcal{Z}(\mathbf{z}, \mathbf{z}') \leq \delta \cdot L_2, \tag{12}$$
$$|f(p_\theta(\mathbf{z})) - f(p_\theta(\mathbf{z}'))| \leq L_3 \cdot d_\mathcal{Z}(\mathbf{z}, \mathbf{z}') \leq \delta \cdot L_3.$$

Thus, the difference between the posterior mean of point within the trust region of length $\delta$ centered at $\mathbf{z}$ and the black-box objective value is upper bounded as follows:

$$
\begin{aligned}
&|f(p_\theta(\mathbf{z}')) - m(\mathbf{z}')| \\
&= |(f(\mathbf{x}) - m(\mathbf{z})) + (f(p_\theta(\mathbf{z})) - f(\mathbf{x})) + (f(p_\theta(\mathbf{z}')) - f(p_\theta(\mathbf{z}))) + (m(\mathbf{z}) - m(\mathbf{z}'))| \\
&\leq |f(\mathbf{x}) - m(\mathbf{z})| + |f(p_\theta(\mathbf{z})) - f(\mathbf{x})| + |f(p_\theta(\mathbf{z}')) - f(p_\theta(\mathbf{z}))| + |m(\mathbf{z}) - m(\mathbf{z}')| \\
&\leq c + \gamma \cdot L_1 + \delta \cdot (L_2 + L_3)
\end{aligned} \tag{13}
$$

$\square$

## B Exploration of Potential-aware Trust Region Anchor Selection

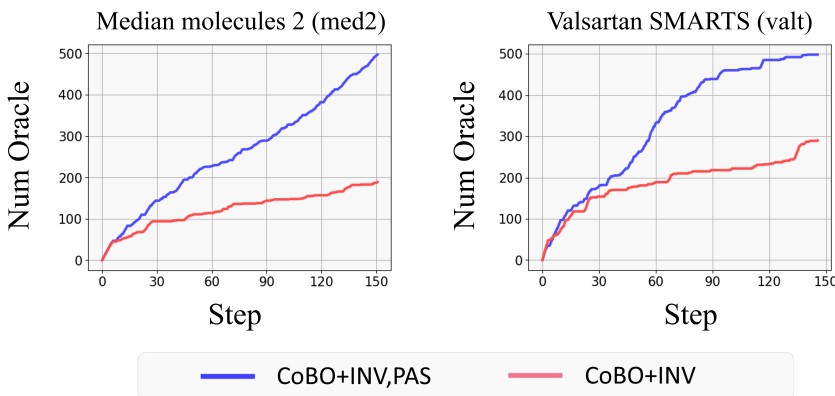

Figure 11: Exploration effectiveness ablation study of the potential aware anchor selection (PAS) on CoBO+InvBO (INV, PAS).

We measure the number of searched unique data samples for each iteration to validate the exploration ability of our potential-aware trust region anchor selection in 4.3. We compare it with the objective score-based anchor selection, which selects the best anchors based on their objective score, used in most prior works [13, 14, 36]. From the figure, our potential-aware anchor selection searches more diverse data compared to the objective score-based anchor selection on both tasks. These results demonstrate that our anchor selection method improves the exploration ability without the loss of the exploitation ability.

# C Dissimilarity in Latent Bayesian Optimization

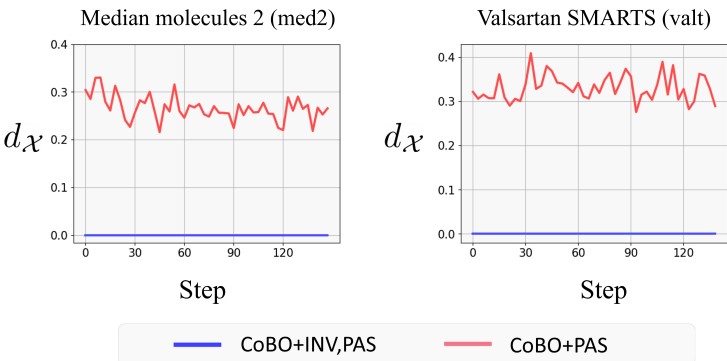

Figure 12: Dissimilarity between $\mathbf{x}^i$ and $p_\theta(\mathbf{z}^i)$ with and without inversion on med2 and valt tasks. The measurement of dissimilarity is the normalized Levenshtein distance between the SELFIES token. (x-axis: number of iterations, y-axis: normalized Levenshtein distance.)

We here provide the comparison of with and without inversion to empirically prove that the inversion generates aligned data. Given a pair of data sample $\mathbf{x}^i$ and its corresponding latent vector $\mathbf{z}^i$, we measure the dissimilarity between the input $\mathbf{x}^i$ and the data $p_\theta(\mathbf{z}^i)$ generated from the decoder. The latent vector generated with the inversion method $\mathbf{z}^i$ is defined as $\mathbf{z}^i = \arg\min_{\mathbf{z} \in \mathcal{Z}} d_\mathcal{X}(\mathbf{x}^i, p_\theta(\mathbf{z}))$ (Section 4.2) The latent vector without inversion method $\mathbf{z}^i = q_\phi(\mathbf{x}^i)$ is constructed by feeding the input $\mathbf{x}^i$ into the encoder $q_\phi$ similar to prior LBO works [11, 17].

Figure 12 shows the dissimilarity comparison results without and with the inversion on med2 and valt tasks. We measure the dissimilarity as the normalized Levenshtein distance between two SELFIES tokens. The x-axis indicates the iteration, and the y-axis indicates the dissimilarity between $\mathbf{x}^i$ and $p_\theta(\mathbf{z}^i)$. From the figure, the inversion achieves zero dissimilarity for every iteration on both tasks, which indicates that the inversion always generates aligned data. On the other hand, BO without the inversion mostly generates misaligned data. These results demonstrate the necessity of the inversion in LBO.

# D Applying PAS to TuRBO on standard BO benchmark

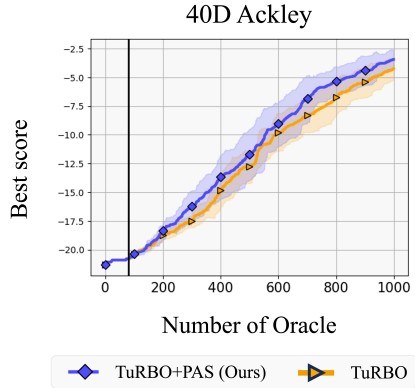

Figure 13: Optimization results of TuRBO and applying PAS to TuRBO on the synthetic Ackley function with 40 dimensions. The lines and ranges indicate the mean and a standard deviation of ten runs with different seeds.

We analyze the effectiveness of PAS along with standard BO approaches. We provide the optimization performance of TuRBO and TuRBO with PAS in the Ackley benchmark function with 40 dimension, with input ranges from [-32.768, 32.768]. Our implementation is based on the codebase of TuRBO [36] provided in the BoTorch [48] tutorial, and we use the RBF kernel with Automatic Relevance Determination (ARD) lengthscale [49]. The number of initial data is 80 and the number of

total oracle calls is 1,000. Figure 13 shows that applying PAS on TuRBO consistently improves the optimization performance during the iteration. This result demonstrates that PAS is also effective in standard BO benchmark tasks.

## E    Experiments of Plug-and-Play

We provide further results of applying our InvBO to previous LBO works, LS-BO, TuRBO-*L*, W-LBO [11], LOL-BO [13], PG-LBO [17] and CoBO [14] on Guacamol and arithmetic fitting tasks. All results are average scores of ten runs under identical settings. Since LS-BO, W-LBO, and PG-LBO are not trust region-based LBO, we report the optimization results applying inversion (INV) only on these baselines. The experimental results are illustrated in Table 2. The table shows that applying our InvBO to all previous LBO works consistently improve the optimization performance in all Guacamol and arithmetic expression fitting tasks.

## F    Experiments on Guacamol Benchmark with Small Budget

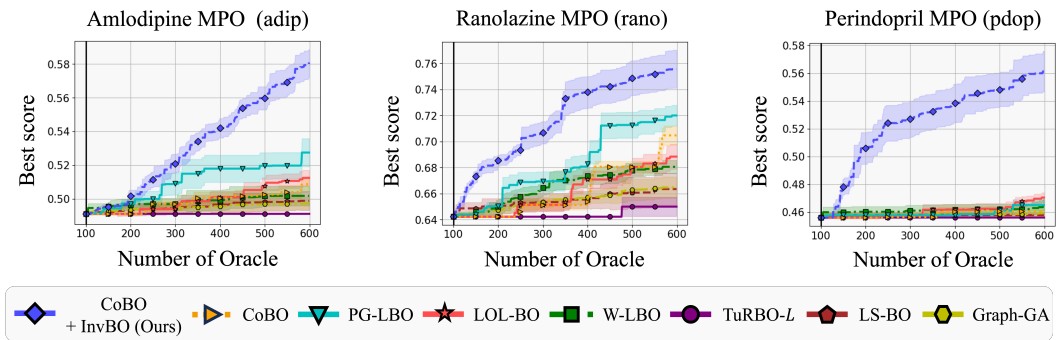

Figure 14: Optimization results on Guacamol benchmark tasks, excluding the tasks in Figure 4. The lines and ranges indicate the average and standard error of ten runs under the same settings. A higher score is a better score.

Figure 14 provides the optimization results on three Guacamol benchmark tasks including pdop, adip, and rano. The y-axis of each subfigure denotes the best-found score, and the x-axis denotes the number of the objective function evaluation. Our InvBO built on the CoBO shows the state-of-the-art performance on all three tasks. In particular, CoBO with our InvBO achieves 0.56 best score in the pdop task while the best scores of all other baselines are under 0.48.

## G    Experiments on Guacamol Benchmark with Large Budget

We provide further optimization results on five Guacamol benchmark tasks, adip, osmb, pdop, rano, and zale under the large budget setting to demonstrate the effectiveness of our InvBO in diverse budget settings, which are used in previous works [13, 14]. The experimental results are illustrated in Figure 15. In the figure, our InvBO to CoBO achieves superior performance on large-budget settings. Specifically, InvBO built on the CoBO showed a large margin from the baseline and achieved a more than 0.9 best score in adip tasks. These results imply that our InvBO is effective in diverse settings and tasks.

## H    Analysis on Recentering Technique

We here provide the details about additional oracle calls in the recentering technique [13, 14]. Firstly, CoBO explicitly mentioned the oracle call for recentering in L8-9 of Algorithm 1 in the paper. Secondly, LOL-BO's oracle calls for recentering can be verified by its official GitHub code. See, L188-228 of `lolbo/lolbo/lolbo.py` and L54 of `lolbo/lolbo/latent_space_objective.py`.

Table 2: Optimization results of applying InvBO or inversion method (INV) to several LBOs on Guacamol benchmark tasks and arithmetic expression task, including the task in Table 1. A higher score is better for all tasks except the arithmetic expression task.

| Task | | Aripiprazole similarity (adip) | | | Median molecules 2 (med2) | | |
|---|---|---|---|---|---|---|---|
| Num Oracle | | 100 | 300 | 500 | 100 | 300 | 500 |
| LBO | | 0.493±0.003 | 0.497±0.005 | 0.499±0.006 | 0.186±0.000 | 0.186±0.000 | 0.186±0.000 |
| LBO + INV | | **0.495±0.002** | **0.501±0.006** | **0.509±0.008** | 0.186±0.000 | **0.188±0.002** | **0.190±0.002** |
| W-LBO | | 0.497±0.003 | 0.499±0.006 | 0.502±0.006 | 0.186±0.000 | 0.186±0.000 | 0.186±0.000 |
| W-LBO + INV | | **0.507±0.001** | **0.523±0.002** | **0.526±0.002** | 0.186±0.000 | **0.206±0.001** | **0.207±0.001** |
| PG-LBO | | 0.497±0.003 | 0.518±0.008 | 0.528±0.008 | 0.186±0.000 | 0.186±0.000 | 0.186±0.000 |
| PG-LBO + INV | | **0.498±0.003** | **0.523±0.006** | **0.558±0.007** | 0.186±0.000 | 0.186±0.000 | **0.191±0.002** |
| TuRBO-L | | 0.491±0.000 | 0.491±0.000 | 0.491±0.000 | 0.186±0.000 | 0.186±0.000 | 0.186±0.000 |
| TuRBO-L + InvBO | | **0.499±0.006** | **0.518±0.008** | **0.541±0.008** | 0.186±0.000 | **0.194±0.002** | **0.202±0.001** |
| LOL-BO | | 0.491±0.000 | 0.501±0.004 | 0.512±0.005 | 0.186±0.000 | 0.186±0.000 | 0.190±0.001 |
| LOL-BO + InvBO | | **0.500±0.002** | **0.545±0.009** | **0.578±0.011** | **0.189±0.002** | **0.204±0.005** | **0.227±0.010** |
| CoBO | | 0.491±0.000 | 0.501±0.004 | 0.509±0.005 | 0.186±0.000 | 0.188±0.002 | 0.191±0.003 |
| CoBO + InvBO | | **0.502±0.004** | **0.542±0.006** | **0.581±0.008** | **0.187±0.001** | **0.203±0.004** | **0.214±0.006** |

| Task | | Osimertinib MPO (osmb) | | | Valsartan SMARTS (valt) | | |
|---|---|---|---|---|---|---|---|
| Num Oracle | | 100 | 300 | 500 | 100 | 300 | 500 |
| LBO | | 0.762±0.000 | 0.763±0.001 | 0.769±0.005 | 0.000±0.000 | 0.000±0.000 | 0.000±0.000 |
| LBO + INV | | **0.767±0.002** | **0.773±0.004** | **0.782±0.004** | 0.000±0.000 | 0.000±0.000 | 0.000±0.000 |
| W-LBO | | 0.762±0.001 | 0.766±0.003 | 0.771±0.004 | 0.000±0.000 | 0.000±0.000 | 0.001±0.001 |
| W-LBO + INV | | **0.770±0.001** | **0.786±0.001** | **0.789±0.001** | 0.000±0.000 | **0.002±0.001** | **0.116±0.010** |
| PG-LBO | | 0.763±0.002 | 0.764±0.002 | 0.771±0.003 | 0.000±0.000 | 0.000±0.000 | 0.000±0.000 |
| PG-LBO + INV | | **0.765±0.003** | **0.784±0.005** | **0.804±0.004** | 0.000±0.000 | 0.000±0.000 | **0.159±0.079** |
| TuRBO-L | | 0.762±0.000 | 0.762±0.000 | 0.762±0.000 | 0.000±0.000 | 0.000±0.000 | 0.000±0.000 |
| TuRBO-L + InvBO | | **0.765±0.002** | **0.785±0.003** | **0.799±0.004** | 0.000±0.000 | **0.024±0.017** | **0.212±0.092** |
| LOL-BO | | 0.762±0.000 | 0.769±0.002 | 0.777±0.003 | 0.000±0.000 | 0.000±0.000 | 0.000±0.000 |
| LOL-BO + InvBO | | **0.775±0.002** | **0.797±0.006** | **0.807±0.005** | 0.000±0.000 | **0.007±0.005** | **0.171±0.039** |
| CoBO | | 0.762±0.000 | 0.763±0.001 | 0.772±0.003 | 0.000±0.000 | 0.000±0.000 | 0.000±0.000 |
| CoBO + InvBO | | **0.769±0.002** | **0.795±0.004** | **0.804±0.004** | 0.000±0.000 | **0.042±0.013** | **0.348±0.107** |

| Task | | Perindopril MPO (pdop) | | | Ranolazine MPO (rano) | | |
|---|---|---|---|---|---|---|---|
| Num Oracle | | 100 | 300 | 500 | 100 | 300 | 500 |
| LBO | | 0.458±0.002 | 0.458±0.002 | 0.458±0.002 | 0.650±0.006 | 0.655±0.007 | 0.664±0.012 |
| LBO + INV | | **0.466±0.007** | **0.466±0.007** | **0.469±0.007** | **0.658±0.007** | **0.669±0.008** | **0.682±0.009** |
| W-LBO | | 0.460±0.004 | 0.462±0.004 | 0.464±0.005 | 0.649±0.006 | 0.674±0.007 | 0.681±0.005 |
| W-LBO + INV | | **0.468±0.001** | **0.478±0.002** | **0.483±0.001** | **0.683±0.002** | **0.700±0.003** | **0.707±0.003** |
| PG-LBO | | 0.458±0.002 | 0.458±0.002 | 0.465±0.004 | 0.650±0.003 | 0.681±0.009 | 0.720±0.008 |
| PG-LBO + INV | | 0.458±0.002 | **0.466±0.005** | **0.490±0.009** | **0.674±0.010** | **0.713±0.013** | **0.729±0.016** |
| TuRBO-L | | 0.456±0.000 | 0.456±0.000 | 0.456±0.000 | 0.642±0.000 | 0.642±0.000 | 0.650±0.007 |
| TuRBO-L + InvBO | | **0.470±0.005** | **0.506±0.008** | **0.534±0.008** | **0.688±0.004** | **0.743±0.011** | **0.791±0.013** |
| LOL-BO | | 0.456±0.000 | 0.462±0.004 | 0.470±0.004 | 0.642±0.000 | 0.671±0.007 | 0.688±0.010 |
| LOL-BO + InvBO | | **0.512±0.017** | **0.546±0.014** | **0.565±0.014** | **0.699±0.004** | **0.762±0.012** | **0.787±0.014** |
| CoBO | | 0.456±0.000 | 0.460±0.004 | 0.461±0.004 | 0.642±0.000 | 0.654±0.006 | 0.705±0.007 |
| CoBO + InvBO | | **0.506±0.011** | **0.538±0.014** | **0.561±0.015** | **0.686±0.004** | **0.738±0.012** | **0.756±0.015** |

| Task | | Zaleplon MPO (zale) | | | Arithmetic expression | | |
|---|---|---|---|---|---|---|---|
| Num Oracle | | 100 | 300 | 500 | 100 | 300 | 500 |
| LBO | | 0.344±0.009 | 0.357±0.010 | 0.368±0.009 | **1.274±0.147** | 0.512±0.046 | 0.425±0.020 |
| LBO + INV | | **0.370±0.015** | **0.387±0.010** | **0.394±0.011** | 1.306±0.128 | **0.506±0.114** | **0.317±0.048** |
| W-LBO | | 0.369±0.014 | 0.390±0.013 | 0.407±0.014 | **1.280±0.169** | 0.686±0.103 | 0.538±0.068 |
| W-LBO + INV | | **0.383±0.003** | **0.416±0.002** | **0.423±0.002** | 1.524±0.008 | **0.466±0.006** | **0.445±0.006** |
| PG-LBO | | 0.369±0.009 | 0.410±0.012 | 0.415±0.009 | **1.290±0.146** | 0.913±0.140 | 0.832±0.146 |
| PG-LBO + INV | | **0.372±0.016** | **0.420±0.007** | **0.452±0.006** | 1.515±0.060 | **0.872±0.107** | **0.595±0.106** |
| TuRBO-L | | 0.327±0.000 | 0.332±0.005 | 0.332±0.005 | 1.424±0.126 | 0.885±0.083 | 0.605±0.094 |
| TuRBO-L + InvBO | | **0.411±0.017** | **0.461±0.015** | **0.475±0.016** | **0.898±0.161** | **0.692±0.139** | **0.374±0.060** |
| LOL-BO | | 0.327±0.000 | 0.398±0.012 | 0.431±0.006 | 0.953±0.109 | 0.578±0.053 | 0.412±0.039 |
| LOL-BO + InvBO | | **0.400±0.011** | **0.456±0.011** | **0.477±0.011** | **0.739±0.142** | **0.510±0.031** | **0.392±0.023** |
| CoBO | | 0.327±0.000 | 0.393±0.014 | 0.421±0.010 | **0.752±0.122** | 0.526±0.057 | 0.391±0.006 |
| CoBO + InvBO | | **0.435±0.014** | **0.495±0.023** | **0.513±0.012** | 0.840±0.173 | **0.513±0.075** | **0.252±0.067** |

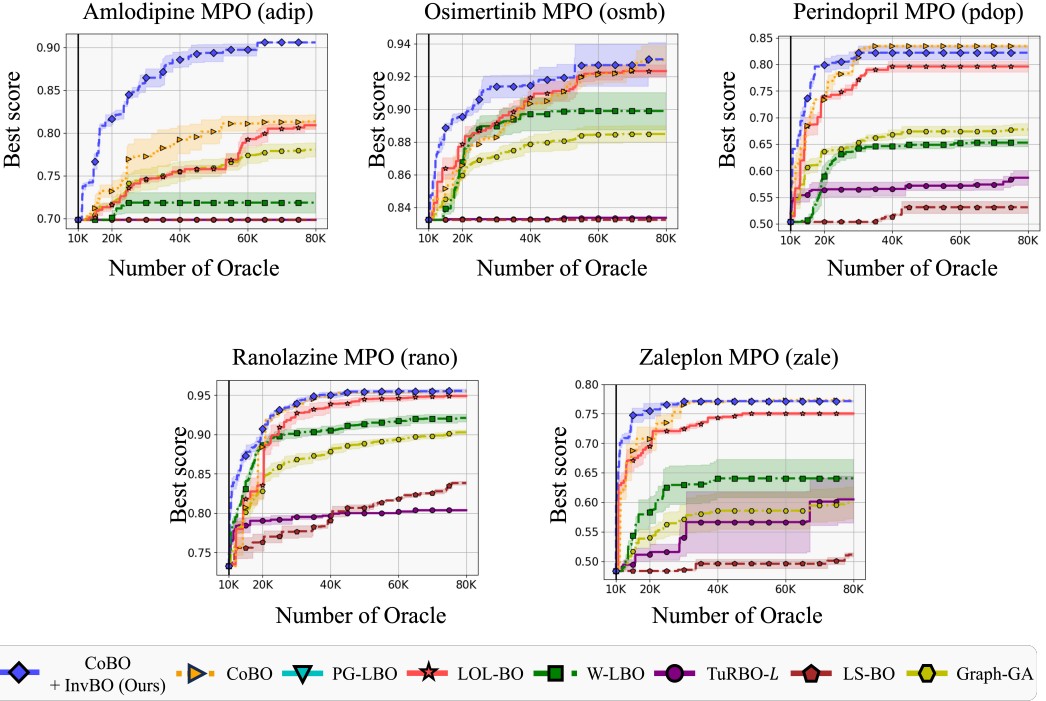

Figure 15: Optimization results on Guacamol task under the large budget setting, excluding the tasks in Figure 5(b). The lines and ranges indicate the average and standard error of five runs under the same settings. A higher score is a better score.

We present the experiments on additional oracle calls in recentering in Figure 16. The figure shows the optimization results of CoBO that use recentering with additional oracle calls and recentering without additional oracle calls on med2 and valt tasks. In both tasks, CoBO that uses the recentering technique without additional oracle calls fails in optimization while consuming additional oracle calls shows progress. These results show that the additional oracle calls are essential in the recentering technique, while the previous works do not explicitly mention it.

## I   Analysis on Misalignment Problem

We here provide the experimental details about the Gaussian process fitting with the encoder triplets and the decoder triplets. We train the surrogate model on 300 train points and predict on 100 test points, all points are randomly sampled. The encoder triplets $\{(\mathbf{x}^i, q_\phi(\mathbf{x}^i), f(\mathbf{x}^i))\}_{i=1}^n$ and the decoder triplets $\{(\mathbf{x}^i, \mathbf{z}_{\text{inv}}^i, f(\mathbf{x}^i))\}_{i=1}^n$ share the same discrete data $\mathbf{x}$, where $\mathbf{x}^i = p_\theta(\mathbf{z}_{\text{inv}}^i)$. The dots in the figure denote the mean predictions of the surrogate model and bars denote 95% confidence intervals. Figure 17 provides the Gaussian process fitting results trained with the encoder triplets (Left) and decoder triplets (Right) that predict the training points on the med2 task. The figure implies that the Gaussian process trained with encoder triplet fails to fit the composite function $f \circ p_\theta$, while trained with decoder triplet emulates the function accurately.

## J   Efficiency Analysis

Although Bayesian optimization assumes the objective function is cost-expensive, it is still important to consider the efficiency of the algorithm. We present an efficiency analysis comparing with baseline methods [11, 13, 14, 17], and TuRBO-$L$, LS-BO. We conducted experiments under the same condition for fair comparison: a single NVIDIA RTX 2080 TI with the CPU of AMD EPYC 7742. Wall-clock time comparison with the baseline model is in Table 3. The table shows that applying our InvBO to CoBO achieves state-of-the-art performance not only under the same oracle calls but also under the same wall-clock time.

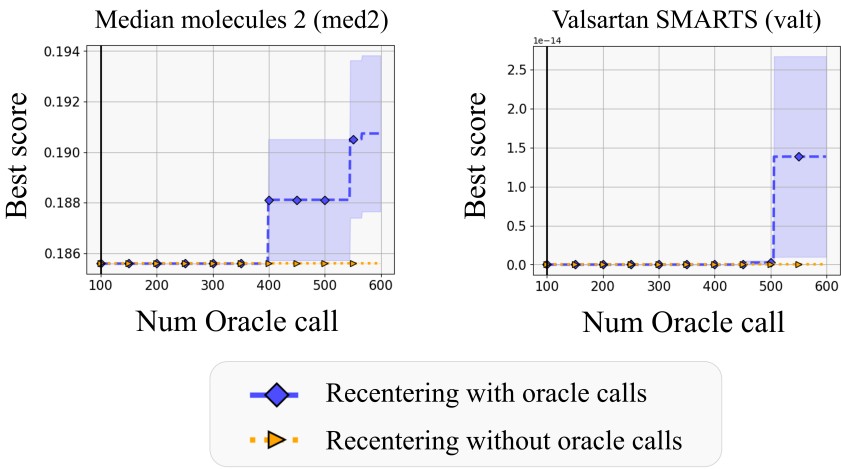

Figure 16: Optimizaiton results on Guacamol tasks, med2 and valt. We compare the CoBO applying InvBO that uses the recentering technique with and without additional oracle calls. The lines and ranges indicate the average and standard error of ten runs under the same settings. A higher score is a better score.

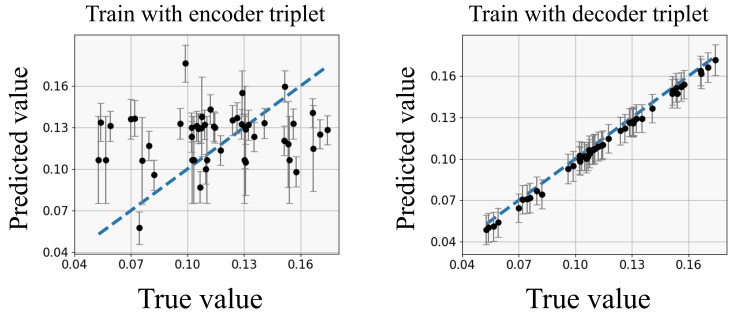

Figure 17: Gaussian process fitting results trained with encoder triplets and decoder triplets. Note that the datapoints are in the training set.

## K   Details of Benchmark Tasks

We demonstrate the effectiveness of InvBO through three optimization benchmarks: Guacamol [43], DRD3, and arithmetic expression fitting task [8, 11, 13, 14]. Guacamol benchmark tasks aim to design molecules with the desired properties, which are scored with a range between 0 to 1. The number of initial points is 100 and the number of oracle calls is 500 in a small-budget setting, and the number of initial points is 10,000 and the number of oracle calls is 70,000 in a large-budget setting. In the DRD3 benchmark task, we aim to design ligands (molecules that specifically bind to a protein) that bind to dopamine receptor D3 (DRD3). We evaluate the docking score of ligands with the Dockstring library [50]. The number of the observed data points is 100 and the number of max oracle calls is 3,000. Lastly, the arithmetic expression fitting task is to generate single-variable expressions that are close to a target expression (e.g., $1/3 + x + \sin(x \times x)$). The number of initial access to oracle is 100 and the number of max oracle calls is 500. All tasks are assumed to be noiseless settings, and we select the initial points randomly from the dataset provided in [14] and the same initial data was used in all tasks.

## L   Details of Baselines

We compare our InvBO with six latent Bayesian Optimization methods: LS-BO, TuRBO [36], W-LBO [11], LOL-BO [13], CoBO [14], and PG-LBO [17], and one graph-based genetic algorithm:

Table 3: Wall-clock time and corresponding best score for each model.

| Task | Model | CoBO+InvBO | CoBO | PG-LBO | LOL-BO | TuRBO | LS-BO | W-LBO |
|------|-------|------------|------|--------|--------|-------|-------|-------|
| med2 | Found Best Score | **0.2143** | 0.1907 | 0.1856 | 0.1904 | 0.1856 | 0.1856 | 0.1856 |
| | Oracle call | 500 | 500 | 500 | 500 | 500 | 500 | 500 |
| | Wall-clock time (s) | 420.12 | 285.02 | 1582.23 | 308.45 | 842.25 | 152.31 | 573.21 |
| | Found Best Score | **0.1938** | 0.1883 | 0.1856 | 0.1856 | 0.1856 | 0.1856 | 0.1856 |
| | Oracle calls | 284 | 314 | 55 | 286 | 182 | 400 | 236 |
| | Wall-clock time (s) | 152.31 | 152.31 | 152.31 | 152.31 | 152.31 | 152.31 | 152.31 |
| valt | Found Best Score | **0.7325** | 0.0000 | 0.0000 | 0.0000 | 0.0000 | 0.0000 | 0.0000 |
| | Oracle calls | 500 | 500 | 500 | 500 | 500 | 500 | 500 |
| | Wall-clock time (s) | 246.12 | 148.23 | 443.65 | 163.67 | 101.58 | 152.96 | 133.48 |
| | Found Best Score | **0.1173** | 0.0000 | 0.0000 | 0.0000 | 0.0000 | 0.0000 | 0.0000 |
| | Oracle calls | 262 | 382 | 184 | 403 | 426 | 500 | 439 |
| | Wall-clock time (s) | 101.58 | 101.58 | 101.58 | 101.58 | 101.58 | 101.58 | 101.58 |

GB-GA [45]. We adopt the standard LBO for LS-BO approach following other work [14] with VAE update during the optimization process. TuRBO leverages trust regions to alleviate the over-exploration. To adapt TuRBO to the latent space, we employ TuRBO-$L$, LS-BO with trust region, following other latent Bayesian optimization baselines [13, 14]. W-LBO weights the data based on importance in order to focus on samples with high objective values. LOL-BO adapts the concept of the trust region to the latent space and proposes VAE learning methods to inject the prior of the sparse GP models for better optimization. CoBO designs novel regularization losses based on the Lipschitz condition to boost the correlation between the distance in the latent space and the distance within the objective function. PG-LBO utilizes unobserved data with a pseudo-labeling technique and integrates Gaussian Process guidance into VAE training to learn a latent space for better optimization. In PG-LBO, we sample the pseudo data by adding the Gaussian noise, and dynamic thresholding.

# M    Implementation Details

Our implementation is based on the codebase of [13]. We use PyTorch[3], BoTorch[4] [48], GPy-Torch[5] [52], and Guacamol[6] software packages. In the Guacamol tasks and DRD3 task [50], we use the SELFIES VAE [13] which is pretrained in an unsupervised manner with 1.27M molecules from Guacamol benchmark. The Grammar VAE [8] is pre-trained with 40K expression data in an unsupervised manner. The dimension of latent space is 256 in the SELFIES VAE and 25 in the Grammar VAE. The size of the data obtained from the acquisition function and $k$ are hyperparameters, which are presented in Table 4.

## M.1    Hyperparameters Setting

The learning rate used in the inversion method is 0.1 in all tasks, as we empirically observe that it always finds the latent vector that generates the target discrete data. The maximum iteration number of the inversion method is 1,000 in all tasks. The rest of the hyperparameters follow the setting used in [14]. Since arithmetic fitting tasks and Guacamol with the small budget tasks use different initial data numbers used in [14], we set the number of top-$k$ data and the number of query points $N_q$ same as DRD3 task, as they use the same number of initial data points.

# N    Pseudocode of Potential-aware Trust Region Anchor Selection

Here, we provide the pseudocode of the potential-aware trust region anchor selection method. We get the max acquisition function value, which is a Thompson Sampling, of each trust region $\mathcal{T}^i$ given a

---

[3]Copyright (c) 2016-Facebook, Inc (Adam Paszke) [51], Licensed under BSD-style license
[4]Copyright (c) Meta Platforms, Inc. and affiliates. Licensed under MIT License
[5]Copyright (c) 2017 Jake Gardner. Licensed under MIT License
[6]Copyright (c) 2020 The Apache Software Foundation, Licensed under the Apache License, Version 2.0.

Table 4: Other hyperparameters for various benchmarks.

| Hyperparameters | Guacamol-small | Guacamol-large | DRD3 | Arithmetic |
|---|---|---|---|---|
| Number of initial data points $|D^0|$ | 100 | 10000 | 100 | 100 |
| Number of query points $N_q$ | 5 | 10 | 5 | 5 |
| Number of top-$k$ data | 50 | 1000 | 10 | 10 |
| VAE update interval $N_{\text{fail}}$ | 10 | 10 | 10 | 10 |
| Max oracle calls | 500 | 70000 | 3000 | 500 |

---

**Algorithm 2** Potential-aware Anchor Selection

---

**Input:** Data history $\{\mathbf{x}^i, \mathbf{z}^i, y^i\}_{i=1}^n$, surrogate model $\mathcal{GP}$

1: **for** $i = 1, 2, ..., n$ **do**
2:     Get a candidate set $Z_{\text{cand}}^i$ with random points in the trust region $\mathcal{T}^i$ around $\mathbf{z}^i$
3:     $\alpha_{\text{pot}}^i = \max\limits_{\mathbf{z} \in Z_{\text{cand}}^i} \hat{f}(\mathbf{z})$ where $\hat{f} \sim \mathcal{GP}\left(\mu(\mathbf{z}), k(\mathbf{z}, \mathbf{z}')\right)$                   ▷ *Eq. (7)*
4: **end for**
5: $Y = \{y^i\}_{i=1}^n$
6: $A = \{\alpha_{\text{pot}}^i\}_{i=1}^n$
7: **for** $i = 1, 2, ..., n$ **do**
8:     $\alpha_{\text{scaled}}^i \leftarrow \text{Calculate}(\alpha_{\text{pot}}^i, Y, A)$                   ▷ *Eq. (8)*
9:     $s^i \leftarrow y^i + \alpha_{\text{scaled}}^i$                   ▷ *Eq. (9)*
10: **end for**
11: $S = \{s^i\}_{i=1}^n$
12: **return** $\mathbf{z}^i$ where $i$ is index of $\max(S)$

---

candidate set $Z_{\text{cand}}^i$ in lines 2-3. After scaling the max acquisition function values, we calculate the final score of each trust region $s^i$ in lines 8-9. Finally, we sort the anchors in the descending order of the final score and select the anchor.

## O    Pseudocode of InvBO Applied to CoBO

We provide the pseudocode of CoBO applying our method, InvBO, which consists of the inversion method and the potential-aware anchor selection. The inversion method is used in lines 1 and 8, and the potential-aware anchor selection method is used in line 11. As in CoBO, we use the subset of data history, which is composed of the top $k$ scored data and the most recently evaluated data during the optimization process in line 4. When we fail $N_{\text{fail}}$ times to update the best score, we train the VAE with CoBO loss $\mathcal{L}_{\text{CoBO}}$ and update the dataset to be aligned using the inversion method. After that, we train the surrogate model with $\mathcal{D}^t$ and negative log-likelihood loss $\mathcal{L}_{\text{surr}}$ except for the first iteration and select the trust region anchor with the proposed potential-aware trust region anchor selection in line 11. We select $N_q$ next query points from the trust region in line 12. Then, we evaluate the next query points with the objective function and update the data history in lines 17, and 18.

**Algorithm 3** InvBO built on CoBO

**Input:** Pretrained encoder $q_\phi$, decoder $p_\theta$, black-box function $f$, surrogate model $g$, acquisition function $\alpha$, oracle budget $T$, latent update interval $N_{\text{fail}}$, number of query point $N_q$, initial data $\mathcal{D}^0 = \left\{(\mathbf{x}^i, y^i)\right\}_{i=1}^n$, learning rate for inversion $\eta$, distance function $d_{\mathcal{X}}$

1: $\mathcal{D}^0 \leftarrow \left\{\left(\mathbf{x}^i, \mathbf{z}^i_{\text{inv}}, y^i\right) \mid \left(\mathbf{x}^i, y^i\right) \in \mathcal{D}^0, \mathbf{z}^i_{\text{inv}} \leftarrow \text{Inversion}\left(\mathbf{x}^i, q_\phi, p_\theta, \eta, d_{\mathcal{X}}\right)\right\}_{i=1}^n$    $\triangleright$ *Algorithm 1*
2: $n_{\text{fail}} \leftarrow 0$
3: **for** $t = 1, 2, ..., T$ **do**
4:      $\mathcal{D}^t \leftarrow \text{CONCAT}\left(\mathcal{D}^{t-1}[-N_q :], topk(\mathcal{D}^{t-1})\right)$
5:      **if** $n_{\text{fail}} \leq N_{\text{fail}}$ **then**
6:          $n_{\text{fail}} \leftarrow 0$
7:          Train $q_\phi$ and $p_\theta$ with $\mathcal{L}_{\text{CoBO}}, \mathcal{D}^t$
8:          $\mathcal{D}^t \leftarrow \left\{\left(\mathbf{x}^i, \mathbf{z}^i_{\text{inv}}, y^i\right) \mid \left(\mathbf{x}^i, \mathbf{z}^i, y^i\right) \in \mathcal{D}^t, \mathbf{z}^i_{\text{inv}} \leftarrow \text{Inversion}\left(\mathbf{x}^i, q_\phi, p_\theta, \eta, d_{\mathcal{X}}\right)\right\}_{i=1}^{|\mathcal{D}^t|}$    $\triangleright$ *Algorithm 1*
9:      **end if**
10:      Train $g$ with $\mathcal{L}_{\text{surr}}, \mathcal{D}^t$ if $t \neq 1$ else $\mathcal{D}^0$
11:      $Z_{\text{anchor}} \leftarrow$ Trust Region Anchor Selection$(\mathcal{D}^t, g)$          $\triangleright$ *Algorithm 2*
12:      $Z_{\text{next}} \leftarrow$ Select $N_q$ query points in trust region centered on $Z_{\text{anchor}}$
13:      $y^* \leftarrow \max_{(\mathbf{x}, \mathbf{z}, y) \in \mathcal{D}^t} y$
14:      **if** $f(p_\theta(\mathbf{z}^i)) \leq y^*, \ \forall \mathbf{z}^i \in Z_{\text{next}}$ **then**
15:          $n_{\text{fail}} \leftarrow n_{\text{fail}} + 1$
16:      **end if**
17:      $\mathcal{D}_{\text{new}} \leftarrow \left\{\left(p_\theta(\mathbf{z}^i), \mathbf{z}^i, f(p_\theta(\mathbf{z}^i))\right) \mid \mathbf{z}^i \in Z_{\text{next}}\right\}_{i=1}^{|Z_{\text{next}}|}$
18:      $\mathcal{D}^t \leftarrow \text{CONCAT}\left(\mathcal{D}^t, \mathcal{D}_{\text{new}}\right)$
19: **end for**
20: $(\mathbf{x}^*, \mathbf{z}^*, y^*) \leftarrow \arg\max_{(\mathbf{x}, \mathbf{z}, y) \in \mathcal{D}^T} y$
21: **return** $\mathbf{x}^*$

