# OpenReview forum: "Inversion-based Latent Bayesian Optimization"
_NeurIPS.cc/2024/Conference — NeurIPS 2024 poster_

### Official Review · Reviewer_tYNa · 2024-07-07

**Soundness:** 2
**Presentation:** 3
**Contribution:** 3
**Rating:** 5
**Confidence:** 4

**Summary:**

This paper proposes Inversion-based Latent Bayesian Optimization (InvBO), a plug-and-play module for latent Bayesian optimization (LBO) methods. The key components of InvBO are: 1) An inversion method to find latent codes z that exactly reconstruct input samples x, addressing misalignment between the latent space and input space. 2) A potential-aware trust region anchor selection method that considers both observed function values and the potential of the trust region to improve optimization. Empirically, InvBO boosts the performance of several existing LBO methods on nine tasks spanning molecule design and arithmetic expression fitting. Theoretically, the authors prove InvBO reduces an upper bound on the surrogate model's prediction error within trust regions.

**Strengths:**

The inversion method is a novel and principled way to address the important problem of misalignment between the input and latent spaces in LBO. The method finds decoder triplets (x, z, y) without additional function evaluations.

Potential-aware trust region anchor selection considers both observed values and the acquisition function when selecting local search regions. This expands on prior methods that only use observed values.

The theoretical result in Proposition 1 provides justification for the inversion method. It shows that minimizing the reconstruction error d(x, p(z)) with inversion reduces an upper bound on the surrogate model's error.

Extensive experiments demonstrate the effectiveness of InvBO. It improves over several strong LBO baselines on molecule design (Guacamol, DRD3) and symbolic regression tasks. Ablations confirm both INV and PAS contribute to performance.

The writing is generally clear and easy to follow. Figures, tables and algorithms complement the main text well. The Related Works section provides helpful context.

**Weaknesses:**

The paper could include more discussion on the limitations and practical considerations of LBO in general. For example, what are the trade-offs compared to standard BO? How much data is needed to train an effective VAE? Some discussion is given but more would help ground the work. What are the real applications of LBO?

Proposition 1 relies on several key assumptions, like Lipschitz continuity of f and small reconstruction error d(x, p(z)). While these enable a clean result, it would help to discuss how realistic the assumptions are and what can go wrong if they are violated.

To validate INV and PAS, it is also important to implement them on LBOs, TuRBO-L, LOL-BO.

The potential-aware anchor selection mainly seems motivated by intuition. A theoretical grounding, even if loose, could strengthen this contribution. For example, can we say anything about the quality of the local optima or the regret?

The improvement on DRD3 and the arithmetic expression tasks (Figure 5a) appears smaller than on Guacamol. It would be good to comment on what makes certain tasks more or less challenging for the method.

**Questions:**

Can you comment more on the limitations of LBO and InvBO for practical applications? For example, what size dataset is needed to train an effective VAE model? What happens if the dataset is small?

Proposition 1 assumes the inversion mapping exists and has a small error. Can you discuss what happens if these do not hold, for example, if the inverse mapping is ill-posed?

Can you provide any theoretical insight into why considering the acquisition function in anchor selection (PAS) helps? For example, can it be related to the quality of local optima or regret bounds?

Why is the improvement on DRD3 and the symbolic regression task smaller than on Guacamol? What properties of these tasks make them challenging?

Can you discuss Validating INV and PAS, how about implementing them on LBOs, TuRBO-L, LOL-BO.

**Limitations:**

Limitations of needing a large dataset to train the VAE

Limitations of the theoretical assumptions, and what happens if they are violated in practice

Including this discussion, even if speculative, would help contextualize the contributions for practitioners. The paper already acknowledges several mathematical assumptions. Expanding this to real-world considerations would be a positive addition.

---

> ### Author Rebuttal · Authors · 2024-08-07
>
> - **[W1, Q1, L1] Discussion on the limitations and practical considerations of LBO.**
>
>     Good suggestion. While standard BO struggles with discrete data, LBO addresses this by mapping discrete data to continuous space. To bridge the gap between the discrete and continuous space, LBO utilizes a Variational AutoEncoder (VAE), the quality of which depends on the number of unlabeled data available. Thus, LBO requires a large amount of high-quality unlabeled data to train an effective VAE.
>
> - **[W2, Q2, L2] Realism and violated situation of assumptions in Proposition 1.**
>
>     Thank you for your valuable question. The Lipschitz continuity assumption for the objective function $f$ is common in Bayesian optimization [1-5] or global optimization [6]. Furthermore, we have shown that the distance $d_{\cal X}(\mathbf x, p_\theta(\mathbf z))$ can be reached to zero through our inversion method in Figure 12. These findings imply that our assumptions in Proposition 1 are highly realistic. Notably, the non-zero values of the distance are the main motivation of InvBO. If the assumptions are violated, GP prediction error increases, resulting in suboptimal optimization performance as we have already shown in Figure 8.
>
>     - Reference
>
>         [1] González, Javier, et al. "Batch Bayesian optimization via local penalization." *Artificial intelligence and statistics*. PMLR, 2016.
>
>         [2] Scarlett, Jonathan. "Tight regret bounds for Bayesian optimization in one dimension." *International Conference on Machine Learning*. PMLR, 2018.
>
>         [3] Hoang, Trong Nghia, et al. "Decentralized high-dimensional Bayesian optimization with factor graphs." *Proceedings of the AAAI Conference on Artificial Intelligence*. Vol. 32. No. 1. 2018.
>
>         [4] Kim, Jungtaek, and Seungjin Choi. "On local optimizers of acquisition functions in bayesian optimization." *Machine Learning and Knowledge Discovery in Databases: European Conference, ECML PKDD 2020, Ghent, Belgium, September 14–18, 2020, Proceedings, Part II*. Springer International Publishing, 2021.
>
>         [5] Lee, Seunghun, et al. "Advancing Bayesian optimization via learning correlated latent space." *Advances in Neural Information Processing Systems* 37 (2023).
>
>         [6] Christodoulos A. Floudas and Panos M. Pardalos, editors. Encyclopedia of Optimization, Second Edition. Springer, 2009.
>
> - **[W3, Q5] Validation of INV and PAS on various LBO methods.**
>
>     We have already provided the experimental results of applying InvBO over mentioned LBOs, TuRBO-L, and LOL-BO in Table 2 of Section D. Table 2 shows that applying our InvBO to previous trust region-based LBOs (TuRBO-L, LOLBO, CoBO) and non-trust region-based LBOs (LBO, W-LBO, PG-LBO) consistently improves the optimization performance in 8 tasks.
>
> - **[W4, Q3, L3] Theoretical grounding for PAS.**
>
>     Thank you for the good suggestion. We also have a deep interest in the theoretical grounding for PAS and recognize its significance. To provide a comprehensive theoretical analysis of PAS, we are actively conducting ongoing research in this area. We anticipate sharing these findings in future work.
>
> - **[W5, Q4] Analysis of performance improvement on DRD3 and Arithmetic expression tasks.**
>
>     In the DRD3 and arithmetic expression tasks, the dissimilarity between $\mathbf{x}$ and $p_\theta(\mathbf{z})$ in previous LBOs without InvBO is relatively small, resulting in less room for performance improvement from inversion compared to other tasks. We conducted additional experiments measuring normalized Levenshtein distance between $\mathbf x$ and $p_\theta(\mathbf z)$ with and without inversion on DRD3 and arithmetic expression tasks in Figure 6. in the provided PDF. Compared to Figure 12 in Section C which has a mean of dissimilarity of 0.3, the mean of dissimilarity of these two tasks is lower than 0.1.

---

### Official Review · Reviewer_nFyz · 2024-07-11

**Soundness:** 4
**Presentation:** 4
**Contribution:** 4
**Rating:** 8
**Confidence:** 5

**Summary:**

&nbsp;

The authors propose two empirical improvements to VAE-based Bayesian optimization methodology. First, the authors propose a correction for the eponymous misalignment problem which they characterize in the paper, the idea being that the latent z corresponded to an encoded x may not correspond to the x' decoded from the same latent z. The correction proposed by the authors is based on inversion. The authors demonstrate that inversion systematically improves a suite of VAE-based Bayesian optimization methods and furthermore, include diagnostic experiments that demonstrate how the Gaussian process surrogate model fit is improved following inversion. Secondly, the authors propose an improvement to trust region-based optimization in the VAE latent space based on using "potential-aware" scoring. Given that the method is broadly applicable and the experimental validation of the approach is rigorous, I recommend acceptance with the following points for the authors to consider. I am ready to increase my score if these points are addressed.

&nbsp;

**Strengths:**

&nbsp;

1. The principle strength of this work is the generality of the approach. Concretely, the authors highlight a systematic problem present in all VAE-based Bayesian optimization architectures and solve it with their inversion approach.

2. The diagnostic experiments on the GP fit provided by the authors provides excellent evidence as to why the inversion approach improves performance, shedding light on the underlying mechanism of the inversion approach.

&nbsp;

**Weaknesses:**

&nbsp;

I highlight below some points of concern with the paper. Of particular note is the justification for the potential-aware method. In terms of the sample-efficiency matters benchmark, I believe this would greatly increase the impact of the paper by showcasing the potential for VAE-based BO methods against other molecule generation approaches in the ML literature. If the authors can address these points I will increase my score.

&nbsp;

__MAJOR POINTS__

&nbsp;

1. In Proposition 1, how important is the assumption that the distance function d_\Chi is bounded on [0, 1]?

2. The Levenshtein distance may not be the most appropriate distance for objects such as molecules which are defined using the SMILES and/or SELFIES syntax. Other distance metrics such as Tanimoto similarity [8, 9] would be more chemically meaningful.

3. I think it would be a great idea for the authors to assess their method on the sample efficiency matters benchmark [13]. While this is not a weakness of the current work per se, I list this here as seeing the performance of this method on the benchmark in comparison to other molecule generation approaches would be very interesting to see and would highly encourage me to increase my score.

4. It would be great if the authors could include the T-LBO method from [14] as a baseline as it would be interesting to understand the effect of metric learning on performance. In particular, metric learning is hypothesized to smoothen the latent function and hence make it easier to fit the GP on the latent points. It would be very interesting to understand the interplay between inversion and metric learning. In other words, does inversion directly solve the problem that metric learning is trying to address? For the arithmetic task, the results from this paper should be directly comparable if the experiment was run under the same settings.

5. For the experiments reported in Section C of the appendix, it would be great if the authors could report the Tanimoto similarity between the molecules that correspond to the SELFIES strings as this would be a much more chemically meaningful measure of similarity relative to Levenshtein distance.

6. For the "potential-aware" method of anchor selection the authors use the acquisition function value is scaled because of the effect of using multiple local GPs to model each trust region?. Do the authors have a motivation for using the sum of the objective function value of the anchor together with the scaled maximum of the Thomson sample as the scoring criterion? Presumably the \alpha term accounts for the local region and the objective function accounts for the known quality of the anchor. Why should this objective be substantially different than taking the maximum of a Thompson sample directly i.e. just keeping the \alpha term?

7. Could the authors provide clear instructions for reproducing the experimental results in the supplied code by means of a README? Currently it is not clear how to reproduce the results.

&nbsp;

__MINOR POINTS__

&nbsp;

1. There are some missing capitalizations in the references section e.g. "Bayesian" and "Gaussian".

2. When mentioning variational autoencoders it would be worth citing the original paper [1].

3. On line 68, it may be worth mentioning that the goal of LBO is to learn a latent space to enable optimization over a continuous space from a discrete or structured input space where "structured" refers to objects such as graphs and images.

4. In the related work section it would be worth mentioning the following works on VAE-based Bayesian optimization [2-7].

5. In Proposition 1, f, m, and the composition function of f and p_\theta are assumed to be L_1, L_2, and L_3 Lipschitz continuous respectively. It may be beneficial to clarify that these functions are not 1-Lipschitz, 2-Lipschitz continuous etc. but rather the Lipschitz constants can be arbitrary.

6. The PyTorch, BoTorch, and GPyTorch papers [10-12] should be cited given that the packages are used.

7. In Section K of the appendix the citation for LS-BO is not given (presumably Tripp et al., NeurIPS 2020).

8. In Figures 5 and 6 the x-axis label, "Number of Oracle" is somewhat confusing. Perhaps "queries" would be more appropriate? Additionally, it would be great if the authors could give the number of random trials for which the error bars are reported?

9. Although it is mentioned in the main text, it might be worth adding the number of trials and the fact that the uncertainty bars are standard errors in the captions for Figures 4, 5, and 6.

10. For Table 1, it would be great if the caption appeared above the table rather than below it.

11. The diagnostic experiment on the misalignment problem in Section H of the appendix is interesting. It would be great to report a quantitative R^2 value in addition to the plots.

12. Line 323, "This indicates that both the uncertainty of the surrogate model and objective function value need to be considered for exploration and exploitation". The same can be said about the acquisition function itself?

13. In Algorithm 2 of Section M of the appendix it would be great to explicitly provide the definition of the Calculate subroutine.

14. In Table 3 of Section I of the Appendix it would be worth stating that the rows are fixed wall clock time and fixed oracle calls respectively.

&nbsp;

__REFERENCES__

&nbsp;

[1] Kingma and Welling, Auto-Encoding Variational Bayes, ICLR, 2014.

[2] Stanton, S., Maddox, W., Gruver, N., Maffettone, P., Delaney, E., Greenside, P. and Wilson, A.G., 2022, June. Accelerating Bayesian optimization for biological sequence design with denoising autoencoders. In International Conference on Machine Learning (pp. 20459-20478). PMLR.

[3] Notin, P., Hernández-Lobato, J.M. and Gal, Y., 2021. Improving black-box optimization in VAE latent space using decoder uncertainty. Advances in Neural Information Processing Systems, 34, pp.802-814.

[4] Lu, X., Gonzalez, J., Dai, Z. and Lawrence, N.D., 2018, Structured variationally auto-encoded optimization. In International conference on machine learning (pp. 3267-3275). PMLR.

[5] Siivola, E., Paleyes, A., González, J., & Vehtari, A. (2021). Good practices for Bayesian optimization of high dimensional structured spaces. Applied AI Letters, 2(2), e24.

[6] Maus, N., Wu, K., Eriksson, D. and Gardner, J., 2023, Discovering Many Diverse Solutions with Bayesian Optimization. In International Conference on Artificial Intelligence and Statistics (pp. 1779-1798). PMLR.

[7] Verma, E., Chakraborty, S. and Griffiths, R.R., 2022. High-Dimensional Bayesian optimization with invariance. In ICML Workshop on Adaptive Experimental Design and Active Learning.

[8] Tanimoto TT (17 Nov 1958). "An Elementary Mathematical theory of Classification and Prediction". Internal IBM Technical Report. 1957 (8?).

[9] Bajusz, D., Rácz, A. and Héberger, K., 2015. Why is Tanimoto index an appropriate choice for fingerprint-based similarity calculations?. Journal of cheminformatics, 7, pp.1-13.

[10] Paszke, A., Gross, S., Massa, F., Lerer, A., Bradbury, J., Chanan, G., Killeen, T., Lin, Z., Gimelshein, N., Antiga, L. and Desmaison, A., 2019. PyTorch: An imperative style, high-performance deep learning library. Advances in neural information processing systems, 32.

[11] Balandat, M., Karrer, B., Jiang, D., Daulton, S., Letham, B., Wilson, A.G. and Bakshy, E., 2020. BoTorch: A framework for efficient Monte-Carlo Bayesian optimization. Advances in neural information processing systems, 33, pp.21524-21538.

[12] Gardner, J., Pleiss, G., Weinberger, K.Q., Bindel, D. and Wilson, A.G., 2018. Gpytorch: Blackbox matrix-matrix Gaussian process inference with GPU acceleration. Advances in neural information processing systems, 31.

[13] Gao, W., Fu, T., Sun, J. and Coley, C., 2022. Sample efficiency matters: a benchmark for practical molecular optimization. Advances in Neural Information Processing Systems, 35, pp.21342-21357.

[14] Grosnit, A., Tutunov, R., Maraval, A.M., Griffiths, R.R., Cowen-Rivers, A.I., Yang, L., Zhu, L., Lyu, W., Chen, Z., Wang, J. and Peters, J., 2021. High-dimensional Bayesian optimisation with variational autoencoders and deep metric learning. arXiv preprint arXiv:2106.03609.

&nbsp;

**Questions:**

&nbsp;

1. In Section L, the authors state that they pretrained the SELFIES VAE with 1.27M molecules from the Guacamol benchmark. Do the authors assess whether the generated molecules are contained within the pre-training dataset?

2. What do the authors think the interplay between metric learning and inversion is?

3. On line 248 what is the particular type of approximation that the authors use for their sparse GP implementation? I note that this is also not mentioned in Section L of the appendix. From the code it would appear to be the Sparse Variational Gaussian Process (SVGP) model.

4. Why are Figures 7 and 16 different?


&nbsp;

**Limitations:**

&nbsp;

1. The use of the Levenshtein distance for diagnosing the effect of inversion on the misalignment problem may not be the best criteria for the molecule experiments. Tanimoto similarity would be more indicative of chemically meaningful differences since it is a distance metric between molecules directly and not their string representation.

2. The aforementioned lack of justification for the potential-aware scoring criterion for the trust regions is an additional limitation for the work in its current form.

3. It would be great if instructions could be included in the codebase to reproduce the results.


&nbsp;

---

> ### Author Rebuttal · Authors · 2024-08-07
>
> - **[W1] Importance of $d_{\cal X}$ bounding assumption in Proposition 1.**
>
>     For analytical convenience, we use the bounding assumption of the distance function.  This is similar to the concept of image normalization, where pixel values are scaled to a specific range (often 0 to 1) to facilitate more effective processing and analysis. We would like to note that this scaling does not affect the generality or applicability of our findings, since the underlying principles remain unchanged regardless of the bound.
>
> - **[W2, L1] Appropriateness of Levenshtein distance for molecules.**
>
>     Thank you for the good suggestion. Since we evaluate the optimization problem in diverse domains, such as molecule and arithmetic expression tasks, we use a more general distance metric, Levenshtein distance, compared to Tanimoto similarity. Our InvBO can utilize any distance function, including Tanimoto similarity. We will include it in the final version.
>
> - **[W3] Assessment of InvBO on the Sample Efficiency Matters Benchmark.**
>
>     Thank you for the suggestion. We conduct additional experiments applying InvBO to LBO in the sample efficiency matters benchmark. We apply InvBO to the SELFIES VAE provided by the benchmark on 7 tasks. The experimental results are in the table below:
>
>     |  | REINVENT SMILES | Graph GA Fragment | SELFEIS VAE - InvBO | SELFIES VAE - LBO |
>     | --- | --- | --- | --- | --- |
>     | amlodipine_mpo | 0.635 $\pm$ 0.035 | 0.661 $\pm$ 0.020 | 0.594 $\pm$ 0.011 | 0.516 $\pm$ 0.005  |
>     | median2 | 0.276 $\pm$ 0.008  | 0.273 $\pm$ 0.009 | 0.236 $\pm$ 0.018 | 0.185 $\pm$ 0.001 |
>     | osimertinib_mpo | 0.837 $\pm$ 0.009 | 0.831 $\pm$ 0.005 | 0.823 $\pm$ 0.007 | 0.765 $\pm$ 0.002 |
>     | perindopril_mpo | 0.537 $\pm$ 0.016 | 0.538 $\pm$ 0.009 | 0.538 $\pm $ 0.004 | 0.429 $\pm$ 0.003 |
>     | ranolazine_mpo | 0.760 $\pm$ 0.009 | 0.728 $\pm$ 0.012 | 0.762 $\pm$ 0.011 | 0.452 $\pm$ 0.025 |
>     | valsartan_smarts | 0.179 $\pm$ 0.358 | 0.000 $\pm$ 0.000 | 0.003 $\pm $ 0.005 | 0.002 $\pm$ 0.003 |
>     | zaleplon_mpo | 0.358 $\pm$ 0.062 | 0.346 $\pm$ 0.032 | 0.384 $\pm$ 0.008 | 0.206$\pm$ 0.015 |
>     | SUM | 3.582 | 3.377 | 3.34 | 2.103 |
>     | Rank | 1 | 2 | 3 | 23 |
>
>     We re-rank the method provided in the sample efficiency matters benchmark on 7 tasks by AUC Top-10 from 5 independent runs. In the table, applying InvBO to SELFIES VAE achieves rank 3 on 7 tasks while vanilla LBO with SELFIES VAE achieves rank 23. These results demonstrate that LBO with InvBO is highly competitive with other non-BO baselines.
>
> - **[W4, Q2] Additional comparison to T-LBO and analysis of the interplay between Inversion and metric learning.**
>
>     We appreciate for valuable suggestion. We provide the optimization results of T-LBO with and without inversion on the arithmetic expression task in Figure 5 in the provided PDF. The metric learning proposed in T-LBO adjusts the latent space to be smooth, making it easier to fit the GP. Meanwhile, inversion constructs an aligned dataset without additional oracle calls, enabling the GP to correctly emulate the objective function. Although these are orthogonal approaches, metric learning helps reduce the $L_3$ constant in Proposition 1, and it is expected to produce a synergistic effect with InvBO. We will include it in the final version.
>
> - **[W5] Tanimoto similarity between $\mathbf x$ and $p_\theta(\mathbf z)$.**
>
>     Thank you for the suggestion. We additionally conducted experiments measuring the Tanimoto similarity between $\mathbf{x}$ and $p_\theta(\mathbf{z})$ in Figure 4 in the provided PDF. Figure 4 shows the Tanimoto similarity comparison results with and without inversion on the med2 and valt tasks. We used rdkit library to measure the Tanimoto similarity. From the figure, the inversion achieves a Tanimoto similarity of 1.0 for every iteration, while CoBO without inversion achieves about 0.8 Tanimoto similarities in both tasks.
>
> - **[W6, L2] Motivation for $\alpha$ normalization in PAS.**
>
>     We normalize the $\alpha$ value to ensure that it has the same influence relative to the anchor point’s objective score $y$. Without scaling, regions with high uncertainty may be overly influenced by the $\alpha$ value compared to the objective score $y$.
>
> - **[W7, L3] Providing clear reproduction instructions.**
>
>     We will publish our code, including clear reproduction instructions in the camera-ready version if the paper gets accepted.
>
> - **[Q1] Assessment of generated molecules against the pre-training dataset.**
>
>
>     |  | med2 | valt |
>     | --- | --- | --- |
>     | Ratio of novel data (%) | 100 $\pm$ 0.00 | 100 $\pm$ 0.00 |
>
>     We additionally conducted experiments to assess whether the generated data are contained within the pre-training Guacamol benchmark dataset. The table above provides the ratio of generated data that is not contained in the pre-training dataset on the med2 and valt tasks across five runs. The table demonstrates that all generated data during the optimization process are not contained in the pre-training dataset.
>
> - **[Q3] Sparse GP approximation method used in InvBO.**
>
>     As shown in our codebase, we use Sparse Variational Gaussian Process (SVGP) for the sparse Gaussian process implementation. We will add this detail in our camera-ready version if the paper gets accepted.
>
> - **[Q4] Discrepancies between Figures 7 and 16.**
>
>     Good observation. Figures 7 and 16 are different because they are conducted on different splits. Figure 7 illustrates the Gaussian process fitting results on the test set, while Figure 16 shows the results on the training set.
>
> - **[Minor points in Weakness]**
>
>     We appreciate your detailed review. We will update the final version considering these minor points.

---

> > ### Comment · Reviewer_nFyz · 2024-08-12
> > **Prepared to Champion the Paper for Acceptance Following the Rebuttal**
> >
> > &nbsp;
> >
> > Many thanks to the authors for their rebuttal. I believe the rebuttal has greatly strengthened the paper on two fronts:
> >
> > 1. The results on the sample efficiency matters benchmark are very impressive. Not only does the inversion scheme systematically improve the performance of the SELFIES VAE-BO approach, but the authors achieve a new SOTA on the zaleplon_mpo problem. This is notable because the SOTA is achieved in competition with many general black-box optimization techniques.
> >
> > 2. The Tanimoto similarity results provide further evidence of the efficacy of the inversion mechanism. Specifically, a Tanimoto similarity of 1 showcases that the inversion method is able to recover the same molecule. In contrast without the inversion mechanism it is not possible to recover the same molecule. This feature of the inversion mechanism is very important from a scientific standpoint whereby chemists may be interests in guarantees on recovery of the same molecule.
> >
> > Given the points above, I am prepared to champion this paper for acceptance. The inversion method highlights an important pathology across all VAE-BO architectures, demonstrates systematic empirical improvement by addressing the pathology, demonstrates the mechanism of the pathology and proposed solution, and lastly, produces a new SOTA on a challenging benchmark featuring many general black-box optimizers.
> >
> > &nbsp;

---

> > > ### Author Response · Authors · 2024-08-13
> > >
> > > We appreciate your thorough review. Considering the feedback, we will update the final version if our paper gets accepted.

---

### Official Review · Reviewer_wVbQ · 2024-07-13

**Soundness:** 4
**Presentation:** 4
**Contribution:** 4
**Rating:** 7
**Confidence:** 4

**Summary:**

The authors propose Inversion-based Latent Bayesian Optimization (InvBO), a novel approach to improve latent space Bayesian optimization (LBO) by introducing two novel components. First, to fix the misalignment problem that typically plagues LBO methods that rely on encoder-decoder models, InvBO introduces an inversion method that can be used to recover the latent code that decodes to a given data point. This allows the misalignment problem to be largely circumvented without the need for additional black-box function evaluations. Second, InvBO proposes a new strategy for the selection of the center of the trust region for trust-region based LBO. In existing LBO methods that use trust regions, the trust region centers are usually chosen to be the latent data point associated with the best objective value observed thus far. InvBO proposes a new method for trust region center selection that encourages selection of trust region centers that give the trust region the highest potential to improve local optimization performance. The authors refer to this method as “potential-aware trust region anchor selection”. By combining these two components (the inversion method and the potential-aware trust region anchor selection method), the authors show that InvBO can be applied to substantially outperform current state-of-the-art LBO methods across nine high-dimensional, discrete optimization tasks. These include some of the most difficult tasks from the GaucaMol benchmark suite of molecule optimization tasks.

**Strengths:**

Originality: InvBO proposes two novel ideas that greatly improve performance of LBO. The inversion method is a very straight-forward and effective means of dealing with the misalignment problem that avoids using extra function evaluations. The potential-aware trust region anchor selection method represents a novel means of selecting trust region centers that improves upon the fairly ad-hoc strategy people have been using of just centering the trust region on the best data point observed so far.
Quality: The paper is well-written and concise. Additionally, the figures and tables are all of good quality - they are both easy to parse and do a nice job of displaying results. Figure 2 does a nice job of illustrating the author’s inversion method.
Clarity: The paper is clear and easy to follow from start to finish. The figures and tables are clear and easy to read. The way the authors motivated, defined, and applied InvBO is clear.
Significance: LBO has emerged as one of the most promising ways to optimize black-box functions defined over discrete, high-dimensional spaces. These discrete, high-dimensional optimization problems are of particular interest to the community because relevant real-world problems, particularly in biological design, are defined over discrete high-dimensional spaces (i.e. the discrete search space of molecules or proteins). The results in this paper show a very substantial performance improvement over state-of-the-art methods in LBO, including on some of the most difficult molecular design benchmark optimization tasks in the popular GuacaMol benchmark suite. This paper therefore represents a significant improvement in our ability to optimize discrete high-dimensional black-box functions and will be of interest to the community.

**Weaknesses:**

Figure 1 colors should be changed to be more friendly to red-green color blind folks.

It would be interesting to see an additional comparison to the Genetic Expert-Guided Learning (GEGL) method for the molecular design tasks in the results section. GEGL is an reinforcement learning (RL) method that obtains state-of-the-art performance across tasks in the GuacaMol benchmark suite (see GuacaMol results in Table 2 of their paper here https://arxiv.org/pdf/2007.04897). While I do think that this additional comparison to GEGL would strengthen the paper, I do not think that it is strictly necessary for this paper to be accepted because RL is an orthogonal method and the results currently in the paper compare to all relevant LBO baselines.

**Questions:**

I am interested in your ideas for future work. Do you have plans for how you might build upon this work and continue to improve methods for LBO?

**Limitations:**

Yes.

---

> ### Author Rebuttal · Authors · 2024-08-07
>
> - **[W1] Modifying the colors of Figure 1 for red-green color blind folks.**
>
>     Thank you for the suggestion. We will modify the colors of Figure 1 for red-green color-blind folks in our camera-ready version if the paper gets accepted.
>
> - **[W2] Additional comparison to GEGL.**
>
>     Thank you for the suggestion. We provide the experimental results of GEGL optimization on adip and osmb tasks in Figure 3 in the provided PDF. We include only a subset of baselines in Figure 3 to enhance the clarity of additional experimental results. While GEGL demonstrates superior optimization performance compared to Graph-GA, CoBO with InvBO still achieves higher optimization performance.
>
> - **[Q1] Future work of InvBO.**
>
>     LBO has become a promising approach for optimizing structured data such as molecules or proteins. However, research on multi-objective Bayesian optimization over latent space has not been fully explored [1, 2]. Here, we propose InvBO for single-objective Bayesian optimization over latent space, but we believe that the misalignment problem also occurs in multi-objective Bayesian optimization over latent space. We will explore the adaptation of InvBO to multi-objective Bayesian optimization over latent space.
>
>     - Reference
>         1. Stanton, Samuel, et al. "Accelerating bayesian optimization for biological sequence design with denoising autoencoders." *International Conference on Machine Learning*. PMLR, 2022.
>         2. Gruver, Nate, et al. "Protein design with guided discrete diffusion." *Advances in neural information processing systems* 37 (2023).

---

> > ### Comment · Reviewer_wVbQ · 2024-08-12
> > **Rebuttal Acknowledgement**
> >
> > I would like to thank authors for their response and addressing the points I raised in my review. I am happy to keep my assessment of their work the same.

---

> > > ### Author Response · Authors · 2024-08-13
> > >
> > > Thank you for the constructive review. We will include your valuable feedback in the final version.

---

### Official Review · Reviewer_8vpX · 2024-07-13

**Soundness:** 4
**Presentation:** 3
**Contribution:** 3
**Rating:** 7
**Confidence:** 4

**Summary:**

This paper identifies and addresses an overlooked issue in several latent space Bayesian optimization methods and proposes a new trust region anchor selection method (PAS) that incorporates the "potential" of a trust region to improve optimization. Specifically, the paper proposes an inversion method to correct the misalignment between the encoding that generated a sample and the resultant encoding of the generated sample. Aligning these representations allows for a better estimate by the surrogate and improves optimization. Further, the trust region anchor selection incorporates both the observed objective value at a given point and the potential for improvement within that trust region (evaluated through Thompson sampling), which similarly improves the optimization.

**Strengths:**

- They identify and address the misalignment problem using an inversion method. Logically, this is a more direct and sample efficient way of doing it compared to previous methods which perform oracle evaluations to score unevaluted points. Additionally, this allows for the GP to produce a better fit of the objective which improves optimization.
- The inversion method is plug-and-play with LOLBO and CoBO, and could possibly be used by other LSBO methods that perform fine-tuning of the VAE.
- The potential aware trust region selection is more flexible than TuRBOs and allows the optimization to revisit previous regions in the optimization trajectory. Their ablations show the effectiveness of this.
- The methods are intuitive and straightforward to implement.
- InvBO performs particularly well over baselines in the low-budget regime.

**Weaknesses:**

- Methodologically the contribution is a bit weak due the fact the proposals here are largely extensions of LOLBO and CoBO.
- The Lipschitz assumption for the VAE decoder and objective function doesn't seem well motivated. This isn't a significant issue with respect to the method, but I do question the relevance of Proposition 1.

**Questions:**

PAS is not specific to LSBO and can be employed wherever TuRBO-derived methods are used. Was this tried on one of the standard BO benchmarks (Robot, Lunar, etc)?

**Limitations:**

The noted limitation of InvBO being sensitive to the quality of the generative model is fair and to be expected.

---

> ### Author Rebuttal · Authors · 2024-08-07
>
> - **[W1] Methodological contribution of InvBO beyond LOLBO and CoBO.**
>
>     Our InvBo can be applied beyond trust region-based LBO methods (e.g., LOLBO and CoBO). We show that each component of our InvBO, such as Inversion and Potential-aware trust-region anchor selection (PAS), can be implemented with diverse BO approaches, as specified below:
>
>     - **Inversion.**
>
>         Our Inversion can be applied to any LBO method, including trust region-based LBO methods (e.g., LOLBO and CoBO). In Table 2 of Section D, we have already shown that the inversion has successfully been adopted to diverse LBO works with and without using trust region.
>
>     - **PAS.**
>
>         Our PAS can be extended to any trust region-based standard BO method (e.g., TurBO). To validate it, we conduct additional experiments applying PAS to TuRBO on standard BO tasks in Rover and Lunar and report the optimization results in Figure 2 of the provided PDF. These experimental results demonstrate that InvBO can be applied not only to TR-based LBO but also to standard TuRBO.
>
> - **[W2] Appropriateness of Lipschitz assumptions in Proposition 1.**
>
>     Thank you for your valuable question. In Proposition 1, we assumed the black box function $f$ and the composite function $f\circ p_\theta$ of the black box function and the decoder of VAE as a Lipschitz continuity function. Assuming the black box function or the objective function as the Lipschitz continuous function is common in Bayesian optimization [1-5] or global optimization [6].
>
>     - Reference
>
>         [1] González, Javier, et al. "Batch Bayesian optimization via local penalization." *Artificial intelligence and statistics*. PMLR, 2016.
>
>         [2] Scarlett, Jonathan. "Tight regret bounds for Bayesian optimization in one dimension." *International Conference on Machine Learning*. PMLR, 2018.
>
>         [3] Hoang, Trong Nghia, et al. "Decentralized high-dimensional Bayesian optimization with factor graphs." *Proceedings of the AAAI Conference on Artificial Intelligence*. Vol. 32. No. 1. 2018.
>
>         [4] Kim, Jungtaek, and Seungjin Choi. "On local optimizers of acquisition functions in bayesian optimization." *Machine Learning and Knowledge Discovery in Databases: European Conference, ECML PKDD 2020, Ghent, Belgium, September 14–18, 2020, Proceedings, Part II*. Springer International Publishing, 2021.
>
>         [5] Lee, Seunghun, et al. "Advancing Bayesian optimization via learning correlated latent space." *Advances in Neural Information Processing Systems* 37 (2023).
>
>         [6] Christodoulos A. Floudas and Panos M. Pardalos, editors. Encyclopedia of Optimization, Second Edition. Springer, 2009.
>
> - **[Q1] Applying PAS to TuRBO on standard BO benchmarks.**
>
>     Thank you for the suggestion. We provide the optimization performance of TuRBO and TuRBO with PAS in two standard BO benchmarks, Rover and Lunar. Our implementation is based on the codebase of TuRBO provided in the BoTorch tutorial and uses the same hyperparameters ($e.g.$, batch size, and number of initial data points) with TuRBO. The experimental results are reported in Figure 2 of the provided PDF. The figure shows that applying PAS on TuRBO consistently improves the optimization performance. These results demonstrate that PAS is also effective in standard BO benchmark tasks.

---

> > ### Comment · Reviewer_8vpX · 2024-08-11
> >
> > Thanks for the response and the application of PAS to Rover and Lunar. With those results and the rebuttal, I think my critique of methodological simplicity is not an appropriate weakness.
> >
> > With respect to W2, I still have doubts that we can safely make the Lipschitz assumption for an arbitrary BB objective that we would be interested in optimizing. Nevertheless, CoBO attempts to optimize the Lipschitz continuity of the decoder-objective composition during end-to-end retraining and whenever the conditions of Eqn. 5 are met within a trust region, Prop. 1 will hold. Its likely that Prop. 1 is more reasonable than I first believed.
> >
> > Given the results of the PAS experiment on Rover and Lunar further thought on W2 I'll update my score accordingly.

---

> ### Author Response · Authors · 2024-08-12
>
> Thank you for your thoughtful consideration and revisiting your initial concerns. We will incorporate your valuable feedback in our final version.

---

### Official Review · Reviewer_k47A · 2024-07-13

**Soundness:** 1
**Presentation:** 2
**Contribution:** 2
**Rating:** 3
**Confidence:** 4

**Summary:**

Latent Bayesian optimization has been tackled in this work. In order to solve an optimization problem on a continuous latent space, it utilizes auto-encoder-based neural networks. In particular, this work attempts to solve a misalignment problem in the latent Bayesian optimization. Some experimental results are demonstrated to validate the method proposed in this work.

**Strengths:**

- Latent Bayesian optimization, which is solved in this paper, is a compelling topic in the Bayesian optimization field.

**Weaknesses:**

- Motivation of this work is weak.
- Thorough analysis on the misalignment problem is not provided.
- Experiments are domain-specific.

**Questions:**

- Does the misalignment problem certainly degrade the performance of Bayesian optimization? Is there any particular evidence?
- I think that the proposed method lets each decision focus on exploitation. What do you think about this issue?
- Equation (4) doesn't seem to be inversion. It just finds the nearest output of the decoder.
- How did you choose the dimensionality of the latent space?
- The proposed method seems to require theoretical analysis.
- Considering the nature of Bayesian optimization, which is to solve black-box optimization, the benchmarks used in this work are too domain-specific to show algorithm's performance. Under the assumption of optimizing a black-box function, the proposed method actively utilizes the information of objective functions. Do you expect that your algorithm works well in more general optimization problems?
- I cannot find the details of the neural architectures used. How did you design such networks?
- Could you elaborate on the description of Figure 3?

**Limitations:**

There are no specific societal limitations of this work.

---

> ### Author Rebuttal · Authors · 2024-08-07
>
> - **[W1] Motivation of InvBO.**
>
>     Other reviewers provided positive comments regarding the motivation of our paper as follows:
>
>     > Reviewer 8vpX: This paper identifies and addresses an overlooked issue in several latent space Bayesian optimization methods.
>     >
>
>     > Reviewer wVbQ: The way the authors motivated, defined, and applied InvBO is clear.
>     >
>
>     > Reviewer nFyz:  The authors highlight a systematic problem present in all VAE-based Bayesian optimization architectures.
>     >
>
>     > Reviewer tYNa: The inversion method is a novel and principled way to address the important problem of misalignment between the input and latent spaces in LBO.
>     >
>
>     Here we clarify the motivation behind our proposed InvBO, which consists of Inversion and PAS.
>
>     - **Motivation of Inversion.**
>
>     LBO suffers from the misalignment problem caused by the reconstruction error of the VAE. Previous works handle this problem with a recentering technique; however, Figure 3 shows that this requires additional oracle calls. This motivated us to design an inversion method, a solution to the misalignment problem that does not require any additional oracle calls.
>
>     - **Motivation of PAS.**
>
>     Most prior trust region-based approaches select the anchor as the current optimal point without considering the potential of the latent vectors within the trust region to improve optimization performance. This prompted us to design a novel anchor selection method that considers the potential of the latent vectors within the trust region.
>
> - **[W2, Q1] Evidence and analysis of misalignment problem.**
>
>     We have already presented evidence that the misalignment problem certainly degrades optimization performance. Figure 12 in Section C shows the discrepancy between $\mathbf{x}$ and $p_\theta(\mathbf{z})$. As shown in Figure 7 from the main paper, this discrepancy leads to the misalignment problem. Furthermore, Figure 8 in the main paper shows that optimization performance is significantly lower with a misalignment problem (yellow line) compared to when the problem is addressed by inversion (blue line).
>
> - **[W3, Q6] Diversity of experimental domains and general optimization ability of InvBO.**
>
>     We already measure the performance of our InvBO on diverse domains such as molecule domains (e.g., Guacamol and DRD3 tasks) and an arithmetic expression fitting task. Figures 4 and 5 in the main paper demonstrate the general optimization ability of our InvBO across various domains and settings.
>
> - **[Q2] Exploration capability of InvBO.**
>
>     InvBO does not make exploitation-centric decisions. In Figure 1 of the provided PDF, we conduct additional experiments measuring the number of unique data searched in each iteration of CoBO with and without InvBO. Figure 1 demonstrates that InvBO does not lose exploration capability compared to CoBO. On the other hand, Figure 4 in the main paper shows that applying InvBO enhances the exploitation ability. These results indicate that InvBO makes decisions while balancing exploitation and exploration.
>
> - **[Q3] Clarification on the appropriateness of the term 'Inversion'.**
>
>     ‘Inversion’ is broadly used to refer to the reverse process of generation and has been widely applied to generative models such as GANs and Diffusion models [1-5]. As mentioned in Section 2.2 of the paper, ‘Inversion’ is a process of finding a latent code that generates the original data $\mathbf{x}$ through a generator $G$. Without loss of generality, we also use the term `Inversion' in the same sense with decoder $p_\theta$ and formally define it in Equation (4).
>
>     - Reference
>         1. Xia, Weihao, et al. "Gan inversion: A survey." TPAMI, 2022.
>         2. Zhu, Jiapeng, et al. "In-domain gan inversion for real image editing." *ECCV,* 2020.
>         3. Wang, Tengfei, et al. "High-fidelity gan inversion for image attribute editing." CVPR, 2022.
>         4. Xu, Yiran, et al. "In-N-Out: Faithful 3D GAN Inversion with Volumetric Decomposition for Face Editing." *CVPR,* 2024.
>         5. Gal, Rinon, et al. "An Image is Worth One Word: Personalizing Text-to-Image Generation using Textual Inversion." *ICLR,* 2023
> - **[Q4, Q7] Details of the dimensionality of latent space and the Variational Autoencoder (VAE) used in experiments.**
>
>     For a fair comparison, we use the same dimensionality of latent space and VAE following baseline works. As we mentioned in the main paper, we use SELFIES VAE [1] for the de novo molecule design tasks (e.g., Guacamol and DRD3) and Grammar VAE [2] for the arithmetic expression fitting task.
>
>     - Paper
>         1. Maus, Natalie, et al. "Local latent space Bayesian optimization over structured inputs." NeurIPS, 2022.
>         2. Kusner, Matt J., et al. "Grammar variational autoencoder." *ICML*, 2017.
> - **[Q5] Theoretical analysis of InvBO.**
>
>     In Proposition 1, we theoretically show that inversion plays a crucial role in minimizing the upper bound of the GP prediction error within the trust region. Figure 9 in the main paper shows the experimental results of GP prediction error within the trust region (left) and the corresponding optimization results (right). These results demonstrate that the inversion method minimizes the GP prediction error and results in the improvement of the optimization process.
>
> - **[Q8] Detailed description of Figure 3.**
>
>     The left figure illustrates the number of oracle calls made by the acquisition function and recentering. The right figure displays the number of best score updates achieved by the acquisition function and recentering. From both figures, the acquisition function updates the best score 5 times within approximately 150 oracle calls, whereas the recentering fails to update the best score with about 350 oracle calls. These results demonstrate that the recentering wastes a huge amount of oracle calls.

---

> ### Comment · Reviewer_k47A · 2024-08-13
>
> Thank you for your response.
>
> > [W2, Q1] Evidence and analysis of misalignment problem
>
> In Figure 12, how does it achieve zero dissimilarity? It seems strange. Is there any test data leakage?
>
> Figures 7 and 8 show that the trained model of the proposed method only works for a single specific domain. Please see the concern below.
>
> > [W3, Q6] Diversity of experimental domains and general optimization ability of InvBO
>
> It is the most serious concern. I think that the authors misunderstood my concern. The proposed algorithm is trained on each domain, which implies that the configuration used in this work cannot be used for unseen domains. Bayesian optimization is black-box optimization, so that an optimization method can solve any problem with a small amount of inductive bias.  The authors tackled known domains accessing a true function.
>
> If the proposed method can be applied in unseen tasks without re-training and with the same configuration, I can say that the proposed method is not domain-specific.
>
> > [Q3] Clarification on the appropriateness of the term 'Inversion'
>
> I don't think your answer resolves my concern. First off, "inversion" in GAN inversion is not matched to "inversion" in this work. While GAN models an implicit distribution, the proposed method is defined on an explicit representation. Moreover, Equation (4) does not align with Figure 2(b).
>
> > [Q4, Q7] Details of the dimensionality of latent space and the Variational Autoencoder (VAE) used in experiments
>
> Did you try to adjust the dimensionality of latent space? How does it impact on performance?
>
> Most of my concerns haven't been resolved. I believe that the current manuscript is not ready to be published in NeurIPS.

---

> > ### Author Response · Authors · 2024-08-14
> >
> > Thank you for the thorough review.
> >
> > - **[Q1 in Comment] Clarification of zero dissimilarity in Figure 12 and test data leakage.**
> >
> >     We do not rely on traditional train or test data splits during the inversion process. Inversion is fundamentally a search algorithm designed to find an **“optimal”** latent code $\mathbf z_{\text{inv}}$ that reconstructs the target data $\mathbf x$, rather than training the model and evaluating on test data. In Figure 12, we measure the dissimilarity between $\mathbf x$ and $p_\theta(\mathbf z)$, denoted as $d_{\cal X}(\mathbf x, p_\theta(\mathbf z))$, with and without our inversion method for all observed data. Our Inversion method is designed to find a latent code $\mathbf z_{\text{inv}}$ that reconstructs the target data $\mathbf x$, by minimizing $d_{\cal X}(\mathbf x, p_\theta(\mathbf z))$. The zero dissimilarity of blue line in Figure 12 demonstrates that the inversion method consistently finds an “**optimal”** latent code $\mathbf z_{\text{inv}}$ that satisfies $d_{\cal X}(\mathbf x, p_\theta(\mathbf z_{\text{inv}}))=0$.
> >
> > - **[Q2 in Comment] Applicability of InvBO in unseen domains.**
> >
> >     While a pre-trained VAE cannot be directly applied to an unseen domain, this limitation pertains to most LBOs rather than our proposed InvBO. InvBO, on the other hand, is designed as a plug-and-play module applicable to any VAE-based LBO. Notably, LBO methods employ a fully unsupervised approach for pre-training VAEs, wherein the objective function is not utilized during the pre-training process. In LBO, we assume the availability of sufficient unlabeled data to pre-train a VAE for any given domain.
> >
> > - **[Q3 in Comment] Clarification on the appropriateness of the term 'Inversion'.**
> >
> >     - **Comparing with GAN inversion.**
> >
> >     Both the inversion process in GANs and in our InvBO aim to find a latent code $\mathbf z_{\text{inv}}$ that reconstructs the target data $\mathbf x$. The distinction between whether a generative model defines the data distribution explicitly or implicitly is irrelevant to the appropriateness of the term `Inversion' in InvBO. In both cases, the fundamental goal remains the same: to find the latent code that generates the target data.
> >
> >     - **Consistency between Equation (4) and Figure 2(b).**
> >
> >     Other reviewers provided positive responses regarding the Figure as follows:
> >
> >     > Reviewer wVbQ: Figure 2 does a nice job of illustrating the author’s inversion method.
> >     >
> >
> >     > Reviewer tYNa: Figures, tables and algorithms complement the main text well.
> >     >
> >
> >     Figure 2(b) shows we can find $\mathbf z_{\text{inv}}$ that generates the target evaluated data $\mathbf x$ by inversion method $i.e.,d_{\cal X}(\mathbf x, p_\theta(\mathbf z_{\text{inv}})) = 0$, illustrating the Equation (4).
> >
> > - **[Q4 in Comment] Dimensionality of latent space.**
> >
> >     We did not adjust the dimensionality of latent space. We follow the dimensionality of latent space used in prior works [1].
> >
> >     - Reference
> >
> >         [1] Lee, Seunghun, et al. "Advancing Bayesian optimization via learning correlated latent space." *Advances in Neural Information Processing Systems* 37 (2023).

---

### Author Rebuttal · Authors · 2024-08-07

Thank you to the reviewers for the thorough feedback on our paper. Based on the reviews, we have organized the key strengths of our paper that reviewers identified:

### **1. Convincing motivation.**

Most of the reviewers (8vpX, wVbQ, nFyz, tYNa) provided positive feedback on our motivation. We address the misalignment problem that has been overlooked by prior LBO works, despite its presence in all VAE-based Bayesian optimization architectures.

### **2. Superior optimization performance.**

To validate the effectiveness of InvBO, we measured the optimization performance on nine different tasks. Reviewers (8vpX, wVbQ) responded positively to the superior optimization performance of InvBO, especially in the low-budget setting.

### **3. Novelty.**

Reviewers highlighted the novelty as a strength of InvBO. InvBO consists of two components: the inversion and the PAS method. (wVbQ, tYNa) The inversion method provides a novel and principled way to address the misalignment problem. (wVbQ) The PAS method improves the trust region anchor selection, which most previous works approached with ad-hoc strategies.

### **4. Generality.**

Reviewers (8vpX, nFyz) appreciated the generality of InvBO. As mentioned in the paper, InvBO is a plug-and-play algorithm compatible with previous LBOs. Furthermore, we provided experimental results showing that PAS can be applied to TuRBO on standard BO tasks (e.g., non-LBO tasks) in Figure 2 of the provided PDF.

### **5. Thorough analysis.**

Reviewers gave positive feedback on our thorough analysis of InvBO. Reviewers (8vpX, tYNa) noted that the ablation studies of InvBO demonstrated the effectiveness of each component. Additionally, the experiments on GP fitting provided strong evidence of the effectiveness of the inversion method (8vpX, nFyz).

---

We appreciate all the reviewers for their thoughtful feedback. We will address all issues raised by the reviewers below.

---

### Decision · Program_Chairs · 2024-09-25

**Decision:**

Accept (poster)

**Comment:**

This paper overall is quite strong, and the reviewers are nearly unanimous about this. The authors have clearly identified a subtle but very important and, in hindsight obvious, problem with existing state of the art methods in latent space BO and their approach to solving it "feels" like the right one. This is clearly borne out in empirical evaluation, and the authors' method is clearly quite strong. It feels like essentially every software implementation of LOL-BO should probably adopt InvBO as an on-by-default feature. The fact that so many oracle calls can be saved by "in model" changes since they aren't being done directly in service of optimization is great.